# *QuantFormer*: Learning to Quantize for Neural Activity Forecasting in Mouse Visual Cortex

## Abstract

Understanding complex animal behaviors hinges on deciphering the neural activity patterns within brain circuits, making the ability to forecast neural activity crucial for developing predictive models of brain dynamics. This capability holds immense value for neuroscience, particularly in applications such as real-time optogenetic interventions. While traditional encoding and decoding methods have been used to map external variables to neural activity and vice versa, they focus on interpreting past data. In contrast, neural forecasting aims to predict future neural activity, presenting a unique and challenging task due to the spatiotemporal sparsity and complex dependencies of neural signals. Existing transformer-based forecasting methods, while effective in many domains, struggle to capture the distinctiveness of neural signals characterized by spatiotemporal sparsity and intricate dependencies. To address this challenge, we here introduce *QuantFormer*, a transformer-based model specifically designed for forecasting neural activity from two-photon calcium imaging data. Unlike conventional regression-based approaches, *QuantFormer* reframes the forecasting task as a classification problem via dynamic signal quantization, enabling more effective learning of sparse neural activation patterns. Additionally, *QuantFormer* tackles the challenge of analyzing multivariate signals from an arbitrary number of neurons by incorporating neuron-specific tokens, allowing scalability across diverse neuronal populations.
Trained with unsupervised quantization on the Allen dataset, *QuantFormer* sets a new benchmark in forecasting mouse visual cortex activity. It demonstrates robust performance and generalization across various stimuli and individuals, paving the way for a foundational model in neural signal prediction.
Source code available at https://anonymous.4open.science/r/iclr2025_quantformer-E568.

## 1 Introduction

Complex animal behavior is believed to stem from the electrical activity of coordinated ensembles of neurons within specific brain circuits (Yuste, 2015; Yuste et al., 2024). For example, during sensory perception (Ohki et al., 2005; 2006) and motor coordination (Harpaz et al., 2014; Omlor et al., 2019; Santos et al., 2015), correlated patterns of electrical activity in groups of neurons are observed in the primary sensory and motor cortices (Chen et al., 2024; Inagaki et al., 2022; Panzeri et al., 2022). These activity patterns are structured both spatially and temporally, meaning different subsets of neurons are activated at distinct times. The patterns are further distinguished based on the sensory stimuli or motor outputs they represent (Kondapavulur et al., 2022; Miller et al., 2014; Rule & O'Leary, 2022). Importantly, the activity at any given moment is influenced by the recent history of the neuronal circuit (Boly et al., 2007; Leinweber et al., 2017; Luczak et al., 2022).

Neuronal activity patterns can be recorded in the intact brain using various methods, including electrophysiological recordings (Jun et al., 2017; Steinmetz et al., 2019) and optical techniques such as two-photon microscopy (Denk et al., 1990; Helmchen & Denk, 2005) combined with fluorescent activity reporters (Chen et al., 2013; Dana et al., 2019). These methods enable high-resolution, in vivo imaging of brain cell activity, allowing researchers to observe coordinated neuronal responses during sensory stimulation and motor execution. For instance, studies have shown specific neuronal ensembles encoding stimulus features and behavior in the sensory cortex (Buetfering et al., 2022; Carrillo-Reid et al., 2019), and in the motor cortex during motor programs (Serradj et al., 2023).

A key challenge in neuroscience is developing predictive models that can forecast neuronal activity in a given brain network based on past observations. This task holds significant scientific value, particularly for online closed-loop experiments, such as optogenetics, where real-time adjustments to experimental conditions can enhance intervention effectiveness. Our approach to **forecasting neural activity** differs fundamentally from traditional encoding and decoding methods. Decoding methods, such as those detailed in Azabou et al. (2023); Ye et al. (2023); Antoniades et al. (2023), focus on mapping internal neural variables (e.g., neural activity) to external variables, such as behavior occurring simultaneously with the neural response or the stimulus that elicited it. On the other hand, encoding methods Turishcheva et al. (2024a;b); Li et al. (2023); Xu et al. (2023a); Sinz et al., aim to map external variables to internal neural activity. In contrast, our goal is to model future neural activity without relying on synchronous data, emphasizing the unique challenge of forecasting rather than decoding past stimuli/behaviors or encoding past activity.

The motivation for predicting neuronal activity stems from its demonstrated effectiveness in investigating the sensorimotor cortex of humans and nonhuman primates (Truccolo et al., 2010). However, the application of neural activity forecasting to guide online optogenetic manipulation represents a novel and original advancement in this field. A key element in achieving this goal is leveraging data that is accessible in real-time scenarios. Traditional methods often employ spiking activity data (Schrimpf et al., 2018; Pei et al., 2021; Turishcheva et al., 2024a), which presents challenges due to limited accuracy of real-time spike inference. We thus shift the focus on raw fluorescence traces that provide a direct measure of neuronal activity, improving the precision of predictions and enabling effective manipulation in real-time experimental settings.

In this paper, we propose *QuantFormer*, a transformer-based model for two-photon calcium imaging forecasting using latent space vector quantization. Our approach reframes the forecasting problem as a classification task through vector quantization, enabling the learning of sparse activation spikes. Posing a regression problem as a classification task, even when the data is implicitly continuous, facilitates sparse coding (as already demonstrated in pixel (Van Den Oord et al., 2016b) and audio (Van Den Oord et al., 2016a) spaces), which is crucial given the relative rarity of neuronal activations.

QuantFormer first learns, in a masked auto-encoding fashion (Devlin et al., 2018; He et al., 2022), to compress input neural signals into a sequence of quantized codes, thus allowing self-supervised training by predicting masked items in the sequence. This strategy facilitates the pre-training of the model for downstream tasks by quantization learning while providing a natural way to approach forecasting as the prediction of masked future codes.

Scalability to an arbitrary number of neurons is achieved by learning and prepending a set of neuron-specific tokens to the input. These tokens empower the model to process data from all available neurons without the need to create one model for each neuron or to increase its complexity through multivariate analysis, thus facilitating the effective learning of neural dynamics.

*QuantFormer* was extensively evaluated on the publicly available Allen dataset (de Vries et al., 2020), the only existing benchmark - to our knowledge - providing raw fluorescence traces, and significantly surpassed other state-of-the-art forecasting methods in predicting both short- and long-term neural activity. Ablation studies also confirm the design choices underlying *QuantFormer*.

In summary, the key contributions of this work include:

- **Forecasting neural responses for optogenetic manipulation**: We propose a novel approach that leverages neural activity predictions to guide optogenetic interventions, a completely original concept in the literature.

- **Reframing forecasting as a classification problem**: By employing vector quantization, Quant-Former learns a discrete representation of neural signals, enabling the use of classification techniques to predict sparse neuronal activations effectively.

- **Handling arbitrary neural populations**: Our model uses neuron-specific tokens to facilitate the analysis of multivariate signals from any number of neurons, ensuring scalability and generalization across individuals and sessions.

- **Establishing a foundation model for mouse visual cortex**: Leveraging unsupervised learning on the Allen dataset, QuantFormer demonstrates robust forecasting across different stimuli, individuals, and experimental conditions, laying the groundwork for future research in neural signal prediction.

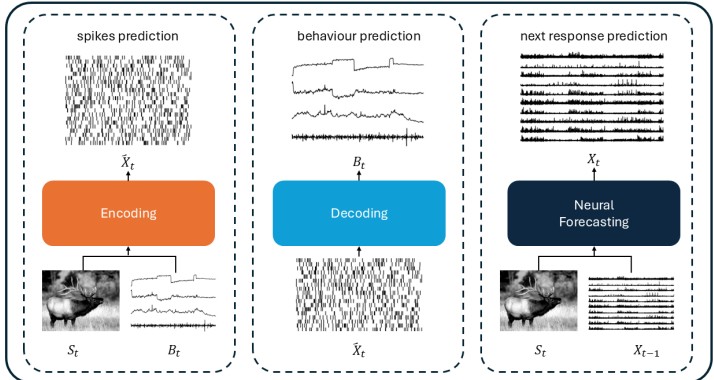

Figure 1: **Comparison of encoding, decoding, and forecasting tasks**. Encoding methods take a stimulus and behavioral variables at time $t$ to predict neural spikes at the same time point. In contrast, decoding methods work do the opposite, using spike responses at time $t$ to predict behavioral variables for that time step. Neural forecasting differs from both, as it uses the stimulus at time $t$ and raw fluorescence traces at time $t-1$ to predict neural responses at time $t$.

## 2 RELATED WORK

This paper introduces *QuantFormer*, a transformer-based method trained using self-supervision for neural forecasting on two-photon calcium imaging data.

In deep learning for two-photon calcium imaging, existing methods have predominantly focused on neuron segmentation (Soltanian-Zadeh et al., 2019; Sità et al., 2022; Bao et al., 2021; Xu et al., 2023b), as well as encoding and decoding tasks.
In particular, *decoding methods* map neural activity (internal variables) to external outcomes like behavior Azabou et al. (2023); Ye et al. (2023); Antoniades et al. (2023). These methods, which use neural activity as input, focus on decoding synchronous patterns, such as behaviors occurring alongside neural responses. However, their goal is not to predict future neural dynamics but to link current neural signals to external events.
*Encoding methods* do the opposite, mapping external variables (e.g., stimuli) to neural activity. Approaches such as Turishcheva et al. (2024a;b); Li et al. (2023); Xu et al. (2023a); Sinz et al. predict neural responses based on stimuli, but often rely on trial-averaged data and are not designed to forecast future neural activity on a single-trial basis without the use of synchronous behaviour variables, which are not accessible in online settings.
In contrast, the task we present in this work, *neural forecasting*, aims to predict future neural activity triggered by external inputs (e.g., visual stimuli) based on the neuron states, i.e., on its past neural data. This is essential for online, closed-loop experiments where forecasting future activity is required to manipulate neurons in real time. The difference between encoding, decoding and neural forecasting tasks is clarified in Fig. 1.

In terms of model architecture, *QuantFormer* is aligned with the recent trend in modeling univariate and multivariate single-dimensional time-series signals through transformers. LogTrans (Li et al., 2019) pioneered the use of transformers in univariate forecasting, utilizing causal convolutions to enhance attention locality. Informer (Zhou et al., 2021) improves efficiency in long-sequence forecasting with sparse attention. PatchTST (Nie et al., 2023) handles multichannel data by processing univariate signals in patches, limiting its ability to capture inter-variable correlations. Crossformer (Zhang & Yan, 2022) applies attention across both time and variable dimensions to exploit multivariate dependencies, with cross-window self-attention capturing long-range relationships. Pyraformer (Liu et al., 2021a) introduces pyramidal attention to represent multiresolution features. FEDformer (Zhou et al., 2022) replaces self-attention with Fourier decomposition and wavelet transforms for handling seasonal data patterns.

*QuantFormer* employs transformers where multivariate signals are handled through prepending tokens that encode specific neurons as well as tokens for stimulus encoding. It employs a pre-training procedure based on reconstructing masked input, in **an autoencoder configuration**, thus leveraging the extensive volumes of unlabeled neural signals from two-photon calcium imaging data in a self-supervised learning setting. This strategy has already demonstrated remarkable results in various research areas, including language modeling (Devlin et al., 2018), audio (Huang et al., 2022), and vision (He et al., 2022; Tong et al., 2022). Additionally, during pre-training, we also learn to quantize in a manner similar to image generation (Esser et al., 2021). However, this quantization approach has not been applied to pose forecasting as a code classification task in the neural signal data domain. Though not directly applied to two-photon calcium imaging, methodologies like BrainLM (Ortega Caro et al., 2023), based on the vanilla transformer, and SwiFT (Kim et al., 2023), which leverages Swin transformers (Liu et al., 2021b) trained on fMRI data, are more closely aligned with our approach, as they perform pre-training through self-learning. Both models are then tuned on downstream tasks, though only BrainLM includes brain state forecasting.

Finally, regarding the data, all the encoding and decoding methods discussed above rely on spiking data (as illustrated in Fig. 1). While spiking data reflects a more processed stage of neural signals, its practical application in online settings is limited due to the challenges of real-time spike inference, which often misses a significant portion of spikes (Huang et al., 2021). Given these limitations, we opted for raw fluorescence traces, which can be captured in real-time and circumvent the pitfalls of spike inference, making them more suitable for online neural forecasting. This decision inherently guided us towards the Allen Visual Coding dataset (de Vries et al., 2020), which provides a comprehensive large-scale benchmark for the mouse visual cortex, encompassing both raw fluorescence traces (unlike other existing benchmarks such as BrainScore (Schrimpf et al., 2018), Neural Latents'21 (Pei et al., 2021), and SENSORIUM (Wang et al., 2023)) and spiking activity.

## 3 METHOD

### 3.1 OVERVIEW

Our approach consists of two distinct training phases: *pre-training* through masked auto-encoding and *downstream training* addressing neural activity classification and forecasting. In the pre-training phase, we train a vector-quantized auto-encoder to reconstruct a sequence of non-overlapping neuronal signal patches - following similar procedures from computer vision (Dosovitskiy et al., 2020; He et al., 2022) - a fraction of which is replaced with a `[MASK]` token. The objective of this task is twofold: first, it encourages the model to learn an expressive and reusable feature representation of neuronal signal for downstream tasks; second, it lays the foundation for its use as a forecasting tool, by using `[MASK]` tokens as placeholders for future signal.

In the downstream phase, we employ the pre-trained encoder to predict neuron activations in response to visual stimuli. As mentioned above, this prediction task can be framed as a time series forecasting task, with the objective of predicting the temporal development of a neuronal response. Alternatively, it can be seen as a classification problem, where an *active* (i.e., neuron activation) or *inactive* (i.e., normal neuron activity) label is associated to the neural signal recorded after stimulus visualization.

### 3.2 PROBLEM FORMULATION

Let $\mathcal{S} = \{s_1, \ldots, s_S\}$ be the set of stimuli to which subjects can be exposed, and let $\mathcal{N} = \{n_1, \ldots, n_N\}$ be the set of neurons under analysis. We define an observation $\mathbf{o} = (\mathbf{x}_b, \mathbf{x}_f, n, s, a)$ to be the set of signals associated to neuron $n \in \mathcal{N}$ when presenting stimulus $s \in \mathcal{S}$: $\mathbf{x}_b \in \mathbb{R}^{L_b}$ is the *baseline activity*, i.e., the neuronal activity *before* the stimulus onset, while $\mathbf{x}_f \in \mathbb{R}^{L_f}$ is the *response activity*, i.e., the neuronal activity *after* the stimulus onset; $a \in \{0, 1\}$ denotes whether neuron $n$ is *active* after the presentation of stimulus $s$, and $L_b$ and $L_f$ denote the temporal length of the different portions in the recorded signals, for a given sample rate $r$. According to Chen et al. (2013), a neuron is marked as *active* ($a = 1$) if the response window has an average gain of 10% over the average baseline luminescence.

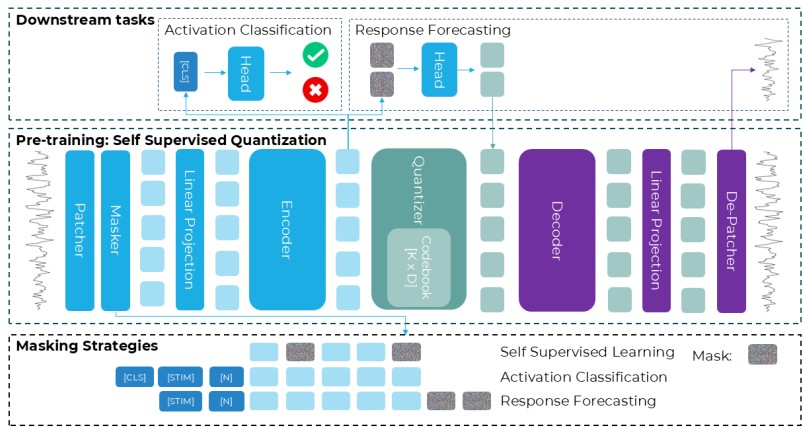

Figure 2: *QuantFormer* **architecture**. During **pre-training** we employ a **self-supervision quantiza-tion strategy** that learns to reconstruct the randomly-masked patches along a quantization scheme. For **response forecasting**, [NEURON] and [STIM] tokens are prepended to the input, and neuronal response patches are masked; the model predicts for the masked patches quantized codes that are converted, through the quantization decoder learned during self-training, to a continuous signal. For **activation classification**, an additional [CLS] token is included in the sequence, and its output embedding is fed to the activation classifier.

The ultimate goal of the proposed approach is to predict neuronal activity in response to a stimulus, by modeling either $p(\mathbf{x}_f|\mathbf{x}_b, n, s)$ (when posing the task as time series forecasting) or $p(a|\mathbf{x}_b, n, s)$ (when posing the task as a classification problem).

## 3.3 PRE-TRAINING STAGE

We pose our self-supervised pre-training as a masked auto-encoding task, with the objective of learning a general representation that models neuronal activity patterns with a view towards response forecasting. In order to make the representation as general as possible (as downstream training will be responsible for specialization), in this stage we aim to reconstruct *the entire signal* for an observation **o**, i.e., the concatenation of $\mathbf{x}_b$ and $\mathbf{x}_f$, while *ignoring both the neuron identity and the presented stimulus*. Formally, let $p(\mathbf{x})$ be the distribution of concatenated baseline and response neuronal signals, with $\mathbf{x} \in \mathbb{R}^{L_b+L_f}$, and let $p(\mathbf{m}|\mathbf{x})$ be a masking function that removes a random portion from **x**. We want to learn a latent representation **z**, from which an estimate of the unmasked signal **x** can be reconstructed as $p(\mathbf{x}|\mathbf{z})$. Following common practices, we model $p(\mathbf{z}|\mathbf{m})$ and $p(\mathbf{x}|\mathbf{z})$ as an encoder-decoder network sharing the latent representation. Additionally, we introduce a quantization layer (Huh et al., 2023) on the latent representation, in order to enforce that the latent representations are mapped to a predefined set of embeddings. While not strictly necessary for pre-training, quantization yields a twofold usefulness for our purposes. First, it enables a categorical representation of neuronal signal components, allowing downstream tasks to pose forecasting as a classification problem rather than a regression one, which has been shown to be easier to optimize (Van Den Oord et al., 2016a). Second, quantization addresses the sparsity of neuronal activations as it forces the model to focus on a limited number of prototypes, encouraging reuse of codes corresponding to common patterns and potentially reducing the impact of over-represented components. We thus define a set of embeddings $E = \{\mathbf{e}_1, \ldots, \mathbf{e}_K\}$, with $\mathbf{e}_i \in \mathbb{R}^d$ and $K$ being the codebook dimension. In this setting, we distinguish between the continuous distribution $p(\mathbf{z}_e|\mathbf{m})$ produced by the encoder network and the categorical distribution $p(\mathbf{z}_q|\mathbf{m})$ obtained after quantization, defined as:

$$p(\mathbf{z}_q = k|\mathbf{m}) = \begin{cases} 1 & \text{with } k = \arg\min_i \|\mathbf{z}_e(\mathbf{m}) - \mathbf{e}_i\|_2 \\ 0 & \text{otherwise} \end{cases} \tag{1}$$

The decoder is then supposed to learn the distribution of $p(\mathbf{x}|\mathbf{z}_q)$. In order to train the entire model, due to impossibility of backpropagating through the quantization operator, a straight-through

estimator (Yin et al., 2019) is employed, directly copying the gradient of the reconstruction loss $\mathcal{L}_{\text{rec}}$ with respect to the quantized representation, $\nabla_{z_q}\mathcal{L}_{\text{rec}}$, to the output of the encoder. We implement $\mathcal{L}_{\text{rec}}$ as the weighted mean squared error between the original unmasked signal $\mathbf{x}$ and the decoder's output, separately taking into account the masked portion $\mathbf{x}_m$ and the unmasked portion $\mathbf{x}_u$:

$$\mathcal{L}_{\text{rec}}(\mathbf{x}, \hat{\mathbf{x}}, a) = (1 + a\beta)\left[\alpha\mathcal{L}_{\text{MSE}}(\mathbf{x}_m, \hat{\mathbf{x}}_m) + \mathcal{L}_{\text{MSE}}(\mathbf{x}_u, \hat{\mathbf{x}}_u)\right], \tag{2}$$

where $\alpha = 2$ emphasizes the importance of predicting masked elements, and $\beta$ is chosen to compensate for the sparsity of neuron activations, by setting its value depending on the ratio between inactive and active neuron observations. Note that this kind of compensation is possible because the model receives an input sequence for a single neuron at a time: multivariate approaches (e.g., Crossformer (Zhang & Yan, 2022)) are unable to balance active and inactive neurons, since a single input packs multiple neurons together. We complement the reconstruction loss with quantization and commitment losses from (Huh et al., 2023), in order to simultaneously train the encoder and learn the codebook.

From an implementation perspective, we employ transformer architectures to model both the encoder and the decoder. Similarly to common approaches in computer vision, we segment the input signal $\mathbf{x}$ into a set of *patches* $\{\mathbf{x}_1, \ldots, \mathbf{x}_P\}$, with $\mathbf{x}_i \in \mathbb{R}^{(L_b+L_f)/P}$ (padding can be applied to make the dimensionality an integer value). A linear projection transforms each patch $\mathbf{x}_i$ into a *token* $\mathbf{t}_i \in \mathbb{R}^d$, which includes positional encoding. The masking function $\mathbf{m}$ replaces a fraction $P_m$ of tokens with a learnable [MASK] token with the same dimensionality as each $\mathbf{t}_i$, producing a masked sequence $\mathbf{m} = \{\mathbf{m}_1, \ldots, \mathbf{m}_P\}$. Following the above formulation:

$$p(\mathbf{m}|\mathbf{x}) = p(\mathbf{m}_1, \ldots, \mathbf{m}_P|\mathbf{t}_1, \ldots, \mathbf{t}_P) = \prod b_i, \tag{3}$$

where each $b_i$ is a Bernoulli random variable with probability $P_m$, such that $\mathbf{m}_i = $ [MASK] if $b_i = 1$, and $\mathbf{m}_i = \mathbf{t}_i$ otherwise. A masked input $\mathbf{m}$ is then fed to the transformer encoder $\mathbf{z}_e$ and quantized into $\mathbf{z}_q$, keeping the same dimensionality as the masked input, i.e., $\mathbf{z}_q \in \mathbb{R}^{P \times d}$. The transformer decoder models $p(\mathbf{x}|\mathbf{z}_q)$ and includes a final projection layer that restores the patch dimensionality from the token representation; merging the resulting patches yields the reconstructed neuronal signal.

## 3.4 DOWNSTREAM TASKS

After pre-training the encoder-decoder network, we employ it for adaptation to specific downstream tasks, namely, *neuron activation prediction* and *response forecasting*.

### 3.4.1 NEURONAL ACTIVATION PREDICTION

Given an observation $\mathbf{o} = (\mathbf{x}_b, \mathbf{x}_s, n, s, a)$, our goal is to predict whether the target neuron responds to the stimulus or not, by modeling $p(a|\mathbf{x}_b, n, s)$. We adapt the pre-trained encoder to this task, with some modifications designed to inject neuron-specific and stimulus-specific knowledge, which was ignored during the self-supervised training. We first introduce a learnable [CLS] token (Devlin et al., 2018; Dosovitskiy et al., 2020), whose representation at the output of the encoder is fed to a linear binary classifier, marking the neuron as *active* or *inactive*. We then define a set of stimulus-specific learnable tokens $\{$[STIM]$_1, \ldots,$ [STIM]$_S\}$, one for each possible stimulus, and a set of neuron-specific learnable tokens $\{$[NEURON]$_1, \ldots,$ [NEURON]$_N\}$, one for each neuron under analysis; all [STIM]$_i$ and [NEURON]$_j$ tokens are $d$-dimensional vectors, i.e., with the same dimension as the encoder input tokens. Given the observation $\mathbf{o} = (\mathbf{x}_b, \mathbf{x}_s, n, s, a)$, we segment and project the baseline signal $\mathbf{x}_b$ into tokens $\{\mathbf{t}_1, \ldots, \mathbf{t}_P\}$, and then feed the encoder network with $\{$[CLS], [NEURON]$_n$, [STIM]$_s$, $\mathbf{t}_1, \ldots, \mathbf{t}_P\}$.

Feeding neuron and stimulus identifiers to the encoder is a key aspect of the approach: since our backbone does not inherently handle multivariate data, we compensate for this lack by providing learnable neuron identifiers, making the model able to learn distinct activation patterns for each neuron; similarly, stimuli identifiers provide a means for the model to discover the specific stimuli a neuron responds to. Also, we only feed the baseline signal $\mathbf{x}_b$ to the encoder, since the response $\mathbf{x}_f$ likely contains neuron activation information, which would defeat the purpose of the classifier.

The transformer-based architecture also allows us to tackle this task in two different training settings: *prompt-tuning* and *fine-tuning*. With prompt-tuning, only soft prompts (i.e., [CLS], all [STIM]$_i$ and all [NEURON]$_j$) can be optimized, while the encoder remains frozen. With fine-tuning, all encoder parameters can be updates. In this task, neither the quantizer nor the decoder are used.

### 3.4.2 NEURONAL RESPONSE FORECASTING

The objective of this task is to model $p(\mathbf{x}_f | \mathbf{x}_b, n, s)$, in order to predict the response time series $\mathbf{x}_f$ from the baseline signal $\mathbf{x}_b$, preceding the stimulus onset. A possible approach to this problem consists in using the pre-trained encoder-decoder network, by masking all input tokens related to the portion of signals to be predicted, and read the forecast signal as the encoder output. However, as mentioned above, the pre-trained model lacks neuron/stimulus specialization, which is needed to handle different neuronal activity dynamics. Moreover, while the pre-trained encoder is trained to capture the underlying patterns of the input data for filling in missing information, this does not necessarily imply the capability to directly predict future values of a time series. To address these issues, we act in two ways: similarly to the previous task, [STIM]$_i$ and [NEURON]$_j$ tokens are added to the beginning of the sequence, to provide the model with specific information; second, rather than fine-tuning the entire model, we append a classification network after the encoder for predicting the codebook indices corresponding to masked tokens only. This approach, besides simplifying the architecture, can be specifically tailored to understand the dynamics of neuronal activity post-stimulus.

Given an input observation $\mathbf{o} = (\mathbf{x}_b, \mathbf{x}_s, n, s, a)$, we convert the baseline signal $\mathbf{x}_b$ into tokens $\{\mathbf{t}_1, \ldots, \mathbf{t}_P\}$, and construct the encoder input as $\{ [\text{NEURON}]_n, [\text{STIM}]_s, \mathbf{t}_1, \ldots, \mathbf{t}_P, [\text{MASK}], \ldots, [\text{MASK}] \}$: the number of [MASK] tokens, $M$, depends on the length $L_f$ of the response signal. [STIM]$_i$ and [NEURON]$_j$ tokens are learned separately from their counterparts in the activation classification downstream task. We denote the output of the encoder corresponding to masked tokens as $\{\mathbf{h}_1, \ldots, \mathbf{h}_M\}$, and feed it to a classification network $\phi$, implemented as a multi-layer perceptron. We compute the set of targets $\{y_1, \ldots, y_M\}$, with $y_i \in \{1, \ldots, K\}$, by feeding the full signal, i.e., the concatenation of $\mathbf{x}_b$ and $\mathbf{x}_f$ to the original pre-trained encoder, reading out the quantization indeces into which the response portion is encoded. We then train the classifier and learn the soft prompts by optimizing the cumulative cross-entropy loss over masked tokens:

$$\mathcal{L}_{\text{rf}} = -\sum_{i=1}^{M} \log \phi \left( \mathbf{h}_i \right)_{y_i} \tag{4}$$

with $\phi \left( \mathbf{h}_i \right)_c$ being the $c$-th component of the predicted class distribution for the $i$-th masked token. At inference time, the predicted codebook indeces replace the masked tokens and the entire sequence is fed to the pre-trained decoder for reconstructing the forecast response. Note that both the codebook and the decoder are frozen at this stage, while the encoder can be frozen too (thus learning soft prompts only during training) or optionally fine-tuned.

## 4 EXPERIMENTAL RESULTS

### 4.1 DATASET

The Allen Brain Observatory Dataset comprises over 1,300 two-photon calcium imaging experiments, organized into more than 400 containers. Each container, representing all the experimental data from a single mouse, consists of three 90-minute sessions foreseeing the administration of multiple stimuli. We selected 11 containers (i.e., mice) previously used in Sità et al. (2022). Each container has at least three complete sessions available. The original dataset includes various types of stimuli: drifting gratings, static gratings, natural scenes, natural movies, locally sparse noise and spontaneous activity (Observatory, 2017). However, we excluded natural movies, as isolating individual neuron responses is challenging, and spontaneous activity, as it is not stimulus-related. Within each session, every stimulus type is presented across three distinct sub-sessions. Each stimulus may be shown once or multiple times. The presentation of a single stimulus, along with its corresponding neural response, is referred to as a trial. In total, we used 236,808 multivariate signals representing neuron responses from the selected mice (additional details in Table A-1).

Moreover, in our classification downstream task, we examine whether there is a response to a given stimulus within a defined response window. Across all mice and their neurons, we identify a total of 2,287,735 positive (*active*) responses, while normal activity (non-responsive, *inactive*) samples amount to approximately 40 million.

To mitigate temporal correlations and prevent overlap between training and test sets, we partition

the data on a per-subsession basis. Specifically, we allocate two subsessions for training and one for testing, with each subsession separated by 10-15 minutes. Furthermore, we ensure that training and testing data are distinctly separated by exposing the mouse to other stimuli during the interim period, thus eliminating potential temporal correlations between signals.

## 4.2 TRAINING PROCEDURE AND METRICS

*QuantFormer* is pre-trained in self-supervision as a masked auto-encoding task through quantization, using data from all subjects. The full model consists of 6 layers and 8 attention heads for both the encoder and the decoder, with a hidden size $d$ of 128 and a mask ratio $P_m$ of 0.2. We train with Adam (Kingma & Ba, 2014) for 50 epochs, a learning rate of $10^{-4}$ and a batch size of 32. In both downstream tasks, we fine-tune the encoder for 100 epochs with a learning rate of $10^{-3}$. The length of baseline and response signals in each observation is respectively 3 seconds and 2 seconds at sample rate $r = 30$, resulting in padded sequence lengths $L_b = 96$ and $L_f = 64$. The number of quantized codes $K$ is set to 32. As we diverge from these values we note that performance decreases in both tasks (see Tables A-2 and A-3 in the *Appendix*). This confirms the sparsity of crucial information in brain signals, which can be encoded with as few as 32 indices (the performance decrease was less sensitive to the dimensionality $d$).

Downstream tasks were conducted separately for each subject and stimulus category, with results reported as mean and standard deviation across all runs. We also evaluate generalization using the *leave-one-category-out* strategy, excluding specific stimulus categories or mice from pre-training and using the excluded data for downstream training. As metrics, we use balanced accuracy, precision, recall, and $F_1$ for classification, and MSE, SMAPE, Pearson correlation and structural similarity index (SSIM) for forecasting.

The selected competitors for our approach, based on code availability and adaptability to the tasks, are Autoformer (Wu et al., 2021), Informer (Zhou et al., 2021), Cross-Former (Zhang & Yan, 2022), and BrainLM (Ortega Caro et al., 2023). We use BrainLM pre-trained on large fMRI data (due to observed similarities between mice and humans (Eppig et al., 2015)), fine-tuned on our data (BrainLM$_{ft}$), and trained from scratch. Additionally, we include a simple LSTM-based baseline, that we empirically found to mostly predict the signal's mean. All experiments are conducted on a workstation with an 8-core CPU, 64GB RAM, and an NVIDIA A6000 GPU (48GB VRAM).

## 4.3 RESULTS

We initially focus on assessing model performance in stimuli response classification; results are shown in Table 1.

Table 1: **Performance on stimuli response classification**. All metrics marked with * have $p \ll 0.01$, while metrics with ** have $p < 0.05$ using one-sided Wilcoxon test.

| Method | Acc ($\uparrow$) | $F_1$ ($\uparrow$) | Prec ($\uparrow$) | Rec ($\uparrow$) |
|---|---|---|---|---|
| LSTM | $61.53 \pm 12.75$* | $31.00 \pm 27.71$* | $40.07 \pm 39.42$* | $35.08 \pm 22.91$* |
| Autoformer | $58.50 \pm 06.11$* | $26.21 \pm 17.06$* | $65.67 \pm 27.18$* | $17.36 \pm 12.47$* |
| Informer | $60.77 \pm 07.13$* | $28.69 \pm 15.53$* | $55.59 \pm 24.82$* | $23.11 \pm 15.69$* |
| BrainLM | $59.66 \pm 13.97$* | $22.71 \pm 32.79$* | $27.25 \pm 39.39$* | $19.56 \pm 02.83$* |
| BrainLM$_{ft}$ | $62.31 \pm 14.67$* | $29.33 \pm 03.48$* | $36.58 \pm 42.11$* | $24.91 \pm 29.69$* |
| Cross-former | $75.51 \pm 04.45$** | $63.89 \pm 07.59$** | $85.71 \pm 03.73$ | $51.49 \pm 09.03$** |
| **QuantFormer** | $\mathbf{77.39} \pm 03.88$ | $\mathbf{66.94} \pm 06.51$ | $\mathbf{85.89} \pm 00.04$ | $\mathbf{55.27} \pm 07.82$ |

*QuantFormer* and Cross-former showcase superior performance compared to other methods, with ours yielding slightly better performance. However, as we discussed earlier, the primary advantage of the quantization strategy lies in its ability to frame a regression task as a classification task for better modeling outliers such as neuron activation. This is demonstrated in Table 2, where we report forecasting metrics, computed on a gradient-based normalization process that scales each signal by dividing it by its accumulated gradient, ensuring that the signals are on a comparable scale based on their overall rate of change (see Sect. III in the *Appendix* for more details). Figure 3 presents qualitative examples of forecast activation predicted by *QuantFormer* and its competitors.

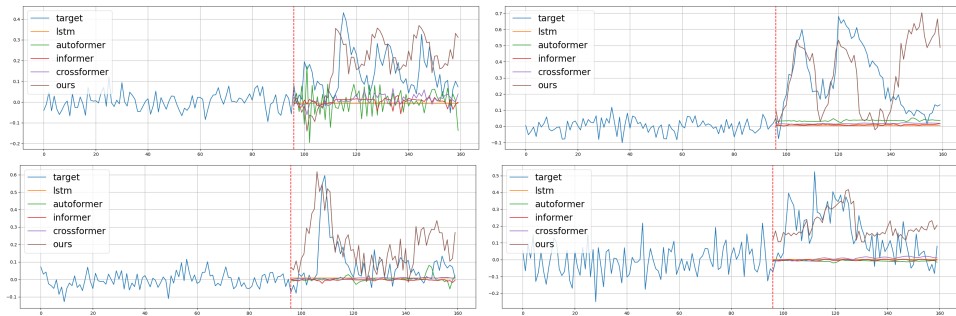

Figure 3: **Qualitative analysis of stimuli response forecasting performance by *QuantFormer* and its competitors**: forecasting examples for each type of stimuli: drifting gratings (top-left), static gratings (top-right), natural scenes (bottom-left) and locally sparse noise (bottom-right). More examples can be found in Section IV of the *Appendix*.

Both quantitative and qualitative results highlight that *QuantFormer* models sparse nature of neural responses better than competitors that predominantly model signals' mean.

Table 2: **Performance on stimuli response forecasting** of *QuantFormer* compared to existing forecasting methods. All metrics marked with * have $p \ll 0.01$, while metrics with ** have $p < 0.05$ using one-sided Wilcoxon test.

| Method | MSE ($\downarrow$) | SMAPE ($\downarrow$) | Corr ($\uparrow$) | SSIM ($\uparrow$) |
|---|---|---|---|---|
| LSTM | $60349.339 \pm$ -* | $0.943 \pm 0.037$* | $0.273 \pm 0.129$* | $0.003 \pm 0.004$* |
| AutoFormer | $19.828 \pm 11.728$* | $0.800 \pm 0.033$* | $0.312 \pm 0.085$* | $0.011 \pm 0.005$* |
| Informer | $0.285 \pm 0.376$** | $0.707 \pm 0.051$* | $0.302 \pm 0.033$* | $0.022 \pm 0.017$* |
| BrainLM | $0.605 \pm 2.852$ | $0.701 \pm 0.158$* | $0.253 \pm 0.125$* | $0.008 \pm 0.034$* |
| BrainLM$_{ft}$ | $0.457 \pm 0.825$ | $0.697 \pm 0.183$* | $0.337 \pm 0.112$** | $0.001 \pm 0.035$* |
| Cross-former | $2.011 \pm 2.749$* | $0.723 \pm 0.062$* | $0.292 \pm 0.087$* | $0.036 \pm 0.020$* |
| **QuantFormer** | $\mathbf{0.247} \pm 0.078$ | $\mathbf{0.656} \pm 0.137$ | $\mathbf{0.338} \pm 0.075$ | $\mathbf{0.069} \pm 0.062$ |

Cross-referencing classification (Table 1) and forecasting performance (Table 2), it becomes apparent that *QuantFormer* excels in both tasks, unlike other methods such as Informer (Zhou et al., 2021) and Cross-former (Zhang & Yan, 2022), which specialize in only one. For instance, while Informer exhibits good forecasting metrics, its classification metrics, especially recall, fall short. This may stem from Informer generating responses with activations surpassing the mean signal, but not reaching the threshold for positive classification. Conversely, Cross-former achieves good classification accuracy but struggles with forecasting, likely due to its tendency to predict constant responses that lead to positive classifications while diverging from actual response patterns.

To substantiate the design choices behind *QuantFormer*, we conduct an ablation study to analyze the importance of different components in the model architectures for classification and forecasting tasks, focusing only on "drifting gratings" stimuli for simplicity. We start by evaluating the performance of our encoder backbone when trained from scratch, using cross-entropy for classification and MSE for forecasting (referred to as $Baseline$ in Table 3). The model is provided with pre-stimulus neuronal activity together with the [STIM] token. We then extend this by prepending the sequence with the [NEURON] token (indicated as $Learnable\ tokens$ in Table 3). Additionally, we evaluate the effects of quantization pre-training on model performance compared to pre-training using a standard auto-encoder scheme without quantization (indicated as $AE$ in Table 3[1]). The results demonstrate that forecasting mostly benefits from embedding quantization.

We also explore pre-training benefits for Informer (Zhou et al., 2021), Autoformer (Wu et al., 2021), and Cross-former (Zhang & Yan, 2022) using quantization and standard auto-encoding. However,

---

[1]Due to space limits, we report only two metrics for classification and two for forecasting.

Table 3: **Ablation study for learnable tokens and quantization on "drifting gratings" stimuli**

| | Classification | | Forecasting | |
|---|---|---|---|---|
| **Method** | **Acc** ($\uparrow$) | $F_1$ ($\uparrow$) | **MSE** ($\downarrow$) | **Corr** ($\uparrow$) |
| Baseline | $75.88 \pm 4.08$ | $64.32 \pm 6.62$ | $0.021 \pm 0.015$ | $0.147 \pm 0.077$ |
| $\hookrightarrow$Learnable tokens | $77.53 \pm 3.89$ | $67.29 \pm 6.39$ | $0.023 \pm 0.018$ | $0.147 \pm 0.055$ |
| $\hookrightarrow$AE | $77.22 \pm 4.25$ | $66.10 \pm 7.41$ | $0.019 \pm 0.014$ | $0.207 \pm 0.082$ |
| $\hookrightarrow$Quantization | $\mathbf{77.66} \pm 3.78$ | $\mathbf{67.42} \pm 6.35$ | $\mathbf{0.016} \pm 0.009$ | $\mathbf{0.252} \pm 0.095$ |

their architectures face two challenges (details in Sect. V of the *Appendix*): 1) combining channel and time information in embeddings creates an information bottleneck, making quantization impractical for temporal patterns; 2) the imbalance between sparse activations and normal signals requires a training strategy targeting individual neurons. Unlike methods that process all neurons simultaneously, *QuantFormer* uses [NEURON] tokens to capture individual neuron dynamics.

We then assess the generalization performance of *QuantFormer* on different subjects and stimuli with a leave-one-out strategy. Table 4 shows that *QuantFormer* generalizes effectively across various scenarios, with performance metrics similar to those in Table 2, underscoring its potential as a foundational model for large-scale studies of the mouse visual cortex.

Table 4: **Generalization performance of *QuantFormer* across subjects and stimuli.**

| | Classification | | Forecasting | |
|---|---|---|---|---|
| | **Acc** ($\uparrow$) | $F_1$ ($\uparrow$) | **MSE** ($\downarrow$) | **Corr** ($\uparrow$) |
| Subjects | $77.32 \pm 4.04$ | $67.18 \pm 6.58$ | $0.367 \pm 0.558$ | $0.344 \pm 0.154$ |
| Stimuli | $76.78 \pm 3.88$ | $67.45 \pm 6.26$ | $0.411 \pm 0.578$ | $0.392 \pm 0.142$ |

In an additional analysis (detailed in the appendix), we examined attention score maps and the latent space of discrete codes and neuron embeddings to understand activation predictions and model interpretability. Attention rollout (Fig. A-8) showed neuron activation predictions are mainly driven by [NEURON] token activity, with pre-stimulus patches and the stimulus token adapting to the specific stimuli. 2D t-SNE on neuron embeddings (Fig. A-10) revealed that the [NEURON] token encodes neuron-specific statistics like activation probability, while 2D t-SNE on the codebook (Fig. A-9) showed discrete codes capture distinct patterns, but their reconstruction is context-dependent, highlighting sequence context in predictions.

## 5 CONCLUSION

We presented *QuantFormer*, a transformer-based model using latent space vector quantization to capture sparse neural activity patterns in two-photon calcium imaging. By framing the regression problem as classification and leveraging unsupervised vector quantization, *QuantFormer* outperforms state-of-the-art methods in response classification and forecasting. Trained and tested on a subset of the Allen dataset, it excels in learning sparse activation spikes and capturing long-term dependencies, making it a versatile and robust tool for understanding neural dynamics.

A possible limitation of *QuantFormer* includes the lack of an inhibition mechanism may lead to sequences of high activation responses, contrary to the typical single activation observed in biological neurons. As future work, *QuantFormer* will be trained on the entire Allen dataset, as well as adapted to spiking neural data (in order to use other existing benchmarks), to enhance generalization capability for creating a foundation model for the mouse visual cortex.

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

# A APPENDIX

## I DATASET DETAILED INFORMATION

The Allen Brain Data Observatory is a resource from the Allen Institute for Brain Science that provides a comprehensive collection of data on the mouse visual cortex. This resource is designed to facilitate research and understanding of brain function, particularly in the context of how sensory information is processed. It contains various types of data regarding the mouse visual cortex ranging from cell connectivity to spontaneous neuronal activity and to stimulus-response data.

For the experiments conducted in this work, as explained in the dataset description subsection, we used the responses to the following four types of stimuli:

- **Drifting gratings:** A full field drifting sinusoidal grating at a spatial frequency of 0.04 cycles/degree was presented at 8 different directions (from 0° to 315°, separated by 45°), at 5 temporal frequencies (1, 2, 4, 8, 15 Hz). Each pattern was shown for 2 seconds, followed by 1 second of a uniform gray background before the next pattern appeared. Also blank sweeps (shown every 20 gratings) are included in this type of stimulus. Each condition (combination of temporal frequency and direction) was presented 15 times across session A. The response time was evaluated on a window of 2 seconds after the stimulus onset.

- **Static gratings:** A full field static sinusoidal grating was presented at 6 different orientations (separated by 30°), 5 spatial frequencies (0.02, 0.04, 0.08, 0.16, 0.32 cycles/degree), and 4 phases (0, 0.25, 0.5, 0.75). Each stimulus was presented for 0.25 seconds, without intergray period. Also, blank sweeps were shown every 25 gratings are included in this type of stimulus. Each condition (combination of spatial frequency, orientation and phase) was presented 50 times across session B. The response time was evaluated on a window of 0.5 seconds after the stimulus onset.

- **Locally Sparse Noise:** This type of stimulus consisted of a 16 x 28 array of pixels, each 4.65 degrees on a side. In each medium gray frame of the stimulus (presented for 0.25 seconds) a small number (11) of pixels were randomly changed to be white or black. 9000 different frames was presented once across session C. The response time was evaluated on a window of 0.5 seconds after the stimulus onset.

- **Natural Scenes:** 118 natural images selected from Berkeley Segmentation Dataset (Martin et al., 2001), van Hateren Natural Image Dataset (van Hateren and van der Schaaf, 1998), McGill Calibrated Colour Image Database (Olmos and Kingdom, 2004) were presented in grayscale for 0.25 seconds each, with no inter-image gray period. Each image was presented 50 times, in random order, and the response period was evaluated in 0.5 seconds after the stimulus onset.

The experimental settings is depicted in Fig. A-1.

Table A-1: **Stimuli administration protocol, dataset information and experiment durations.** Window refers to the length of data (seconds after the administration of the corresponding stimulus) considered for response forecasting. Duration is the time in minutes needed for executing a downstream training epoch for the corresponding stimulus type. The average number of neurons per mouse is 241, with a standard deviation of 63.

| | Acquisition protocol | | | Dataset information | | Duration |
|---|---|---|---|---|---|---|
| **Stimuli type** | **# instances** | **# trials** | **window (s)** | **# mice** | **# signals** | **# time (m)** |
| Drifting gratings | 41 | 628 | 2 | 11 | 6.908 | 0.43 |
| Static gratings | 120 | 6000 | 0.5 | 11 | 66.000 | 2.4 |
| Locally sparse noise | 9000 | 9000 | 0.5 | 11 | 99.000 | 2.04 |
| Natural scenes | 118 | 5900 | 0.5 | 11 | 64.900 | 1.2 |
| Total | 9.279 | 21.528 | – | 11 | 236.808 | 6.07 |

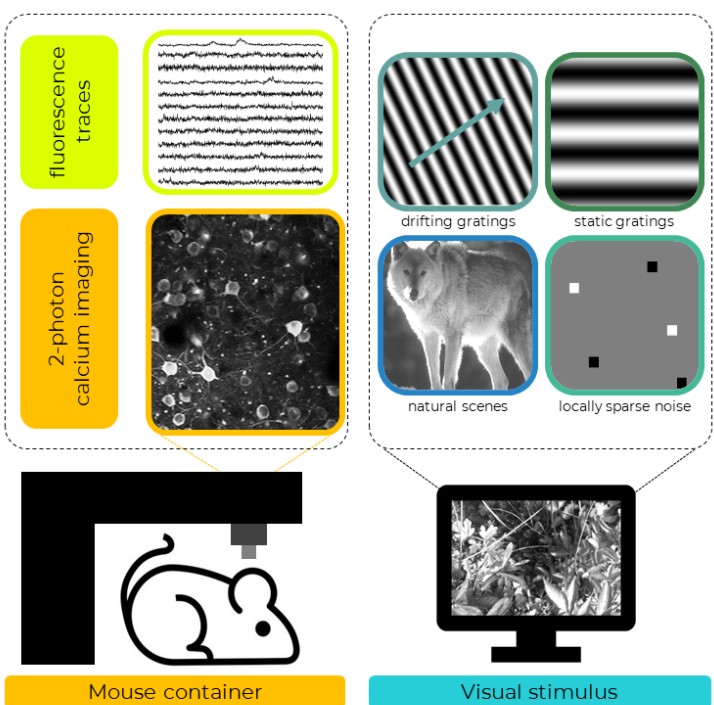

Figure A-1: The Allen dataset. Fluorescence time series are extracted from the two-photon calcium images (*Left*). Examples of the stimuli used (*Right*).

The 11 container ids used for the experiments in this work are: 511507650, 511510667, 511510675, 511510699, 511510718, 511510779, 511510855, 511510989, 526481129, 536323956 and 543677425.

## II HYPERPARAMETER SEARCH FOR QUANTIZATION AND EMBEDDING DIMENSIONALITY

In order to determine the optimal values for the number of quantization indices $(K)$ and embedding dimensionality $(d)$, shared by both the quantized codes and the transformer models, we conduct an exploratory hyperparameter tuning on the responses to "drifting gratings" stimuli only. Such choice was made because this stimuli category needs less time for complete training sessions. First, we fix the value of $d$ to 128 and perform classification and forecasting experiments varying the value of $K$. Our model achieved the best correlation score for a value of $K$ equal to 32. Afterwards, we repeated the same experiments using that number of quantized vectors and we varied the value of parameter $d$ instead.

Table A-2: $K$ **and** $d$ **parameter value search for forecasting.**

| | Forecasting | | | | |
|---|---|---|---|---|---|
| **Value** | **MSE** ($\downarrow$) | **MAE** ($\downarrow$) | **SMAPE** ($\downarrow$) | **Corr** ($\uparrow$) | **SSIM** ($\uparrow$) |
| **Quantization indexes $K$ with $d$ = 128** | | | | | |
| $K = 4$ | $0.028 \pm 0.014$ | $0.132 \pm 0.032$ | $0.744 \pm 0.049$ | $0.161 \pm 0.050$ | $0.010 \pm 0.028$ |
| $K = 8$ | $0.033 \pm 0.006$ | $0.161 \pm 0.019$ | $0.673 \pm 0.174$ | $0.201 \pm 0.065$ | $0.060 \pm 0.0730$ |
| $K = 16$ | $0.015 \pm 0.008$ | $0.081 \pm 0.024$ | $0.647 \pm 0.063$ | $0.219 \pm 0.072$ | $0.091 \pm 0.058$ |
| $K = 32$ | $0.026 \pm 0.005$ | $0.128 \pm 0.015$ | $0.641 \pm 0.154$ | $0.257 \pm 0.077$ | $0.090 \pm 0.086$ |
| $K = 64$ | $0.032 \pm 0.071$ | $0.136 \pm 0.035$ | $0.637 \pm 0.105$ | $0.218 \pm 0.081$ | $0.077 \pm 0.073$ |
| $K = 128$ | $0.040 \pm 0.040$ | $0.141 \pm 0.076$ | $0.631 \pm 0.108$ | $0.221 \pm 0.074$ | $0.076 \pm 0.072$ |
| $K = 256$ | $0.028 \pm 0.015$ | $0.118 \pm 0.041$ | $0.712 \pm 0.103$ | $0.149 \pm 0.043$ | $0.035 \pm 0.051$ |
| $K = 512$ | $0.027 \pm 0.016$ | $0.115 \pm 0.043$ | $0.769 \pm 0.105$ | $0.168 \pm 0.077$ | $0.014 \pm 0.053$ |
| **Embedding dimensionality $d$ with $K$ = 32** | | | | | |
| $d = 64$ | $0.031 \pm 0.01$ | $0.152 \pm 0.028$ | $0.662 \pm 0.18$ | $0.157 \pm 0.01$ | $0.061 \pm 0.06$ |
| $d = 128$ | $0.026 \pm 0.005$ | $0.128 \pm 0.015$ | $0.641 \pm 0.154$ | $0.257 \pm 0.077$ | $0.090 \pm 0.086$ |
| $d = 256$ | $0.051 \pm 0.008$ | $0.179 \pm 0.016$ | $0.764 \pm 0.062$ | $0.234 \pm 0.080$ | $0.016 \pm 0.029$ |
| $d = 512$ | $0.027 \pm 0.028$ | $0.113 \pm 0.064$ | $0.751 \pm 0.111$ | $0.134 \pm 0.020$ | $0.015 \pm 0.052$ |

Table A-3: **Classification performance for varying values of $K$ and $d$.**

| | Classification | |
|---|---|---|
| **Value** | **Acc** ($\uparrow$) | $F_1$ ($\uparrow$) |
| **Quantization indexes $K$ with $d$ = 128** | | |
| $K = 4$ | $76.76 \pm 4.83$ | $66.54 \pm 8.80$ |
| $K = 8$ | $76.80 \pm 4.34$ | $66.57 \pm 8.04$ |
| $K = 16$ | $77.24 \pm 4.72$ | $67.04 \pm 8.37$ |
| $K = 32$ | $77.96 \pm 4.33$ | $66.06 \pm 8.32$ |
| $K = 64$ | $77.45 \pm 4.62$ | $65.70 \pm 7.32$ |
| $K = 128$ | $77.17 \pm 4.92$ | $66.74 \pm 8.31$ |
| $K = 256$ | $77.04 \pm 4.76$ | $66.80 \pm 7.67$ |
| $K = 512$ | $76.90 \pm 4.99$ | $66.63 \pm 8.08$ |
| **Embedding dimensionality $d$ with $K$ = 32** | | |
| $d = 64$ | $76.86 \pm 4.40$ | $66.63 \pm 7.91$ |
| $d = 128$ | $77.96 \pm 4.33$ | $66.06 \pm 8.32$ |
| $d = 256$ | $77.19 \pm 4.66$ | $66.67 \pm 7.99$ |
| $d = 512$ | $64.70 \pm 14.06$ | $37.02 \pm 34.74$ |

The optimal values for $K$ and $d$ were decided by the highest value of Pearson correlation obtained in the downstream task of forecasting (Table A-2, best correlation obtained for values $K = 32$ and $d = 128$). Table A-3, instead, shows the performance obtained in the classification downstream task for varying values of $K$ and $d$.

## III  FORECASTING METRICS FOR UN-NORMALIZED SIGNALS

Table A-4 presents forecasting metrics without normalization, where a basic mean signal baseline yields among the highest performance. However, regression metrics on un-normalized signals, given their sparse nature, does not accurately reflect the true forecasting capabilities of tested models. This motivates our normalization method, which normalizes signals dividing them by the sum of their absolute derivatives, emphasizing the rate of change. This approach highlights true forecasting capabilities and ensures that mean-baseline performance sets the lowest boundary (e.g., $\inf$ for MSE, MAE), penalizing models that predict around the average.

Table A-4: **Regression metrics on stimuli response forecasting using un-normalized responses.**

| Method | MSE ($\downarrow$) | MAE ($\downarrow$) | SMAPE ($\downarrow$) | Corr ($\uparrow$) | SSIM ($\uparrow$) |
|---|---|---|---|---|---|
| Baseline | $0.095 \pm 0.341$ | $0.058 \pm 0.008$ | $0.829 \pm 0.009$ | $0.335 \pm 0.002$ | $0.122 \pm 0.031$ |
| LSTM | $0.093 \pm 0.395$ | $0.505 \pm 0.007$ | $0.883 \pm 0.062$ | $0.252 \pm 0.322$ | $0.246 \pm 0.026$ |
| Autoformer | $0.098 \pm 0.123$ | $0.074 \pm 0.022$ | $0.062 \pm 0.025$ | $0.118 \pm 0.011$ | $0.077 \pm 0.037$ |
| Informer | $0.097 \pm 0.379$ | $0.062 \pm 0.010$ | $0.857 \pm 0.049$ | $0.118 \pm 0.011$ | $0.098 \pm 0.032$ |
| BrainLM | $0.103 \pm 0.388$ | $0.057 \pm 0.008$ | $0.902 \pm 0.049$ | $0.106 \pm 0.006$ | $0.111 \pm 0.031$ |
| BrainLM$_{ft}$ | $0.132 \pm 0.451$ | $0.057 \pm 0.008$ | $0.858 \pm 0.057$ | $0.107 \pm 0.073$ | $0.098 \pm 0.042$ |
| Cross-former | $0.301 \pm 1.210$ | $0.060 \pm 0.009$ | $0.771 \pm 0.039$ | $0.138 \pm 0.032$ | $0.096 \pm 0.032$ |
| **QuantFormer** | $0.445 \pm 1.230$ | $0.236 \pm 0.106$ | $1.55 \pm 0.082$ | $0.138 \pm 0.017$ | $0.015 \pm 0.022$ |

## IV  RESPONSE FORECASTING EXAMPLES

Due to space limitations in the main paper here we report more examples of response forecasts of the tested models to all four categories of stimuli (*Natural scenes* in Figure A-2, *Drifting gratings* in Figure A-3, *Static gratings* in Figure A-4 and *Locally sparse noise* in Figure A-5). All examples showcase the superior capability of *QuantFormer* to model neuron activation w.r.t. competitors.

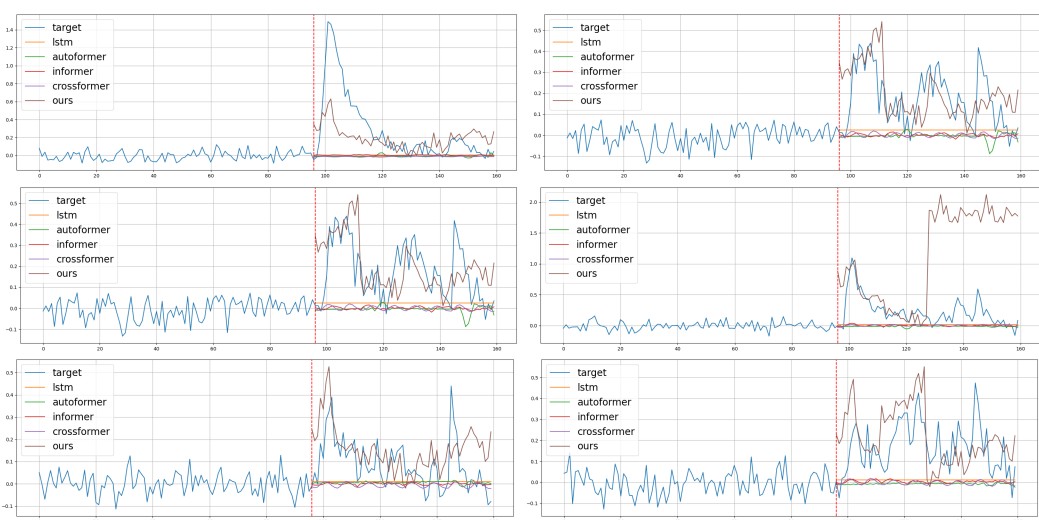

Figure A-2: **Examples of response forecasting by *QuantFormer* and its competitors on natural scenes**.

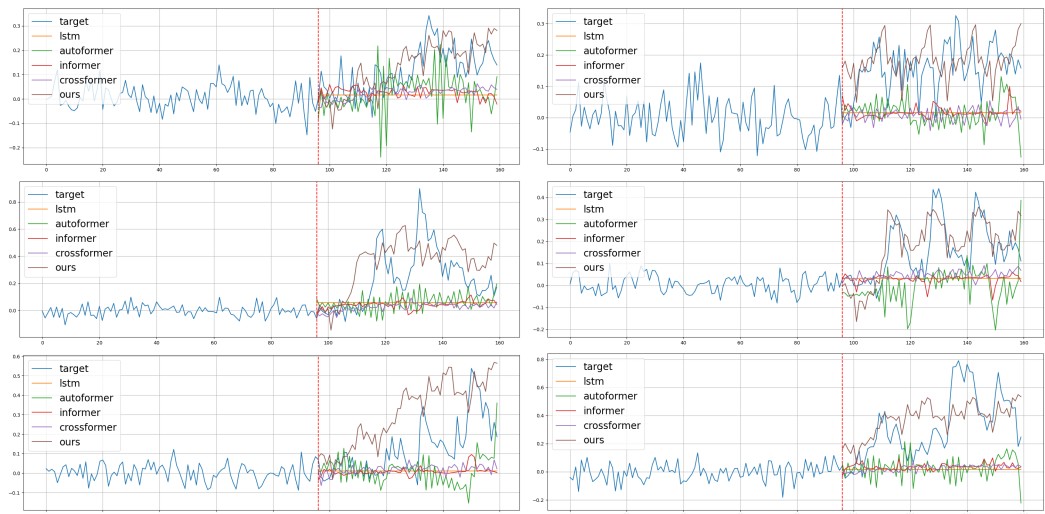

Figure A-3: **Examples of response forecasting by *QuantFormer* and its competitors on drifting gratings**.

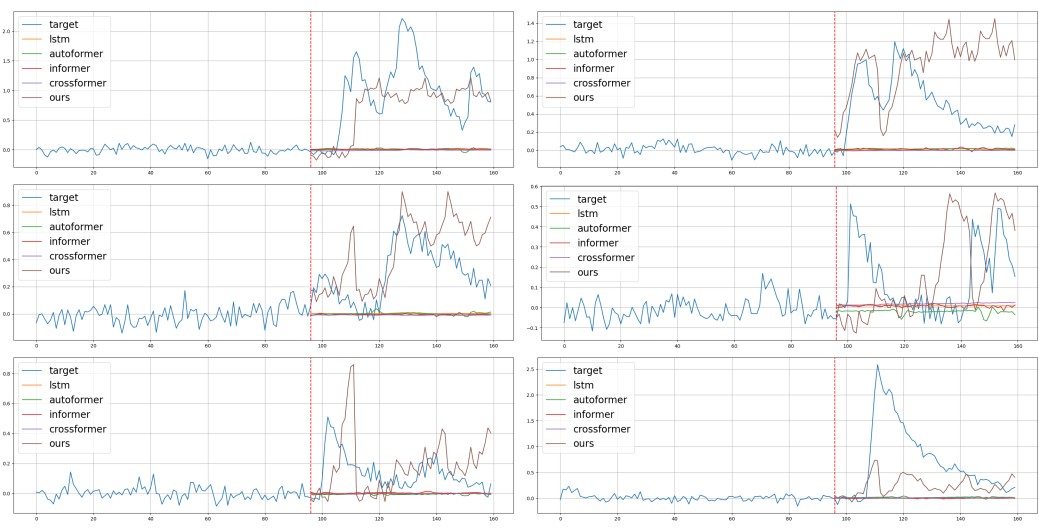

Figure A-4: **Examples of response forecasting by *QuantFormer* and its competitors on static gratings**.

## V APPLICATION OF SELF-SUPERVISED QUANTIZATION ON COMPETITORS

One might question why our pre-training and quantization strategy was not applied to other methods, especially those based on transformer architectures. The primary reason lies in the substantial modifications required to integrate auto-encoding pre-training and quantization into these approaches.

Firstly, quantization is infeasible for Informer (Zhou et al., 2021) and Autoformer (Wu et al., 2021), due to their reliance on embedding layers along the channel dimension, whereas our method embeds temporally patched data. The goal of quantization is to derive robust temporal representations and patterns. Encoding channel combinations with single codes would create an information bottleneck, emphasizing channel patterns over temporal ones.

Secondly, quantization cannot be directly applied to Crossformer (Zhang & Yan, 2022). Although Crossformer performs patching and embedding both channel-wise and temporally, it introduces a two-stage attention mechanism across time and channels. Theoretically, quantization could be

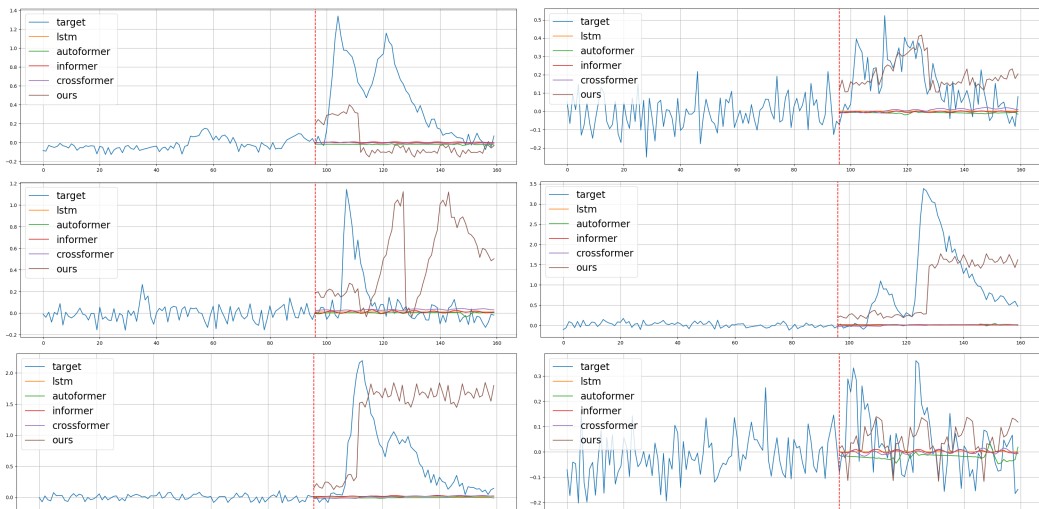

Figure A-5: **Examples of response forecasting by *QuantFormer* and its competitors on locally sparse noise**.

implemented; however, pre-training constraints prevent shuffling, altering, or discarding channels. With only 10% of neurons active per trial, this causes an imbalance during pre-training, leading the quantizer to optimize losses using a limited number of samples for the actual activation. This results in limited number of quantization codes (3) that mostly describe normal activity signals (the majority in the training data), thus leading to a high quantization error, as shown in Fig. A-6. Our approach mitigates this by allowing the exclusion of non-active neurons to maintain data balance during pre-training, thus obtaining a much lower quantization error, as shown in Fig. A-7.

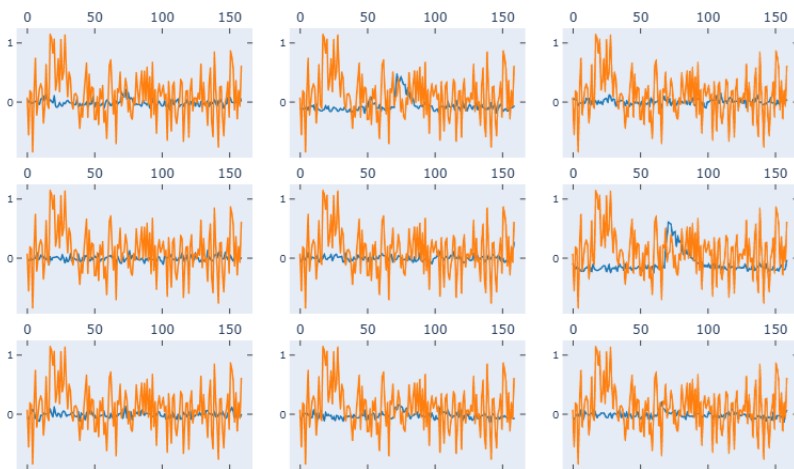

Figure A-6: **Cross-former quantization failure.** In blue, the target signals, while in orange the predicted responses when using quantization.

Furthermore, pretraining itself is problematic for similar reasons. Different containers possess unique channels, necessitating significant alterations to existing methods for effective pretraining. For Autoformer and Informer, each container and experiment would require a dedicated embedding layer to map input channel dimensions into a unified latent space. For Crossformer, introducing a pad token and padding mask might make pretraining feasible, but there would be no consistency in channel order across different containers and experiments. This inconsistency would result in channel attention learning non-generalizable dependencies. Even within a single container, such as a mouse, the number of channels and their order vary across experiments.

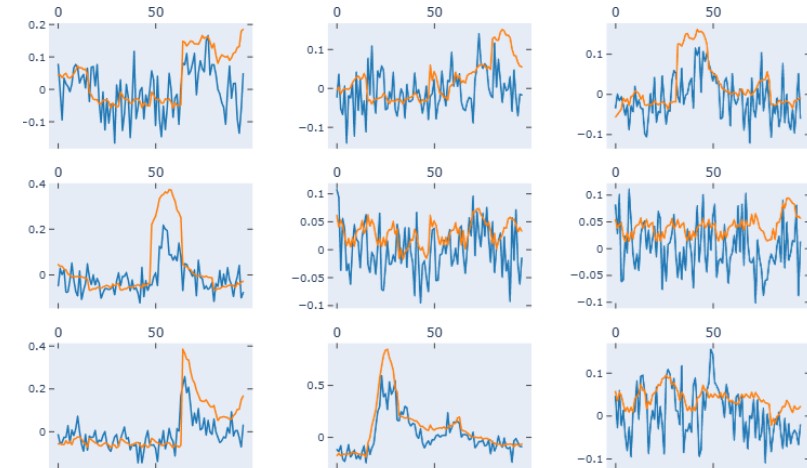

Figure A-7: *QuantFormer* **quantization performance.** In blue, the target signals, while in orange the predicted responses when using quantization.

Thus, our strategy is more appropriate for pretraining, given the inherent challenges and limitations of adapting other methods for this purpose.

# VI    ATTENTION MAPS

We here present in Fig. A-8 attention score maps computed through attention-rollout on *QuantFormer* for neuron activation prediction for all the four types of stimuli: drifting gratings, static gratings, natural scenes, and locally-sparse noise. These maps reveal that [NEURON] token activity predominantly influences predictions, followed by pre-stimulus patches and stimulus token, with the model adapting pre-stimulus information based on the specific stimuli delivered. These attention maps show distinct activation patterns requiring further investigation by neuroscientists.

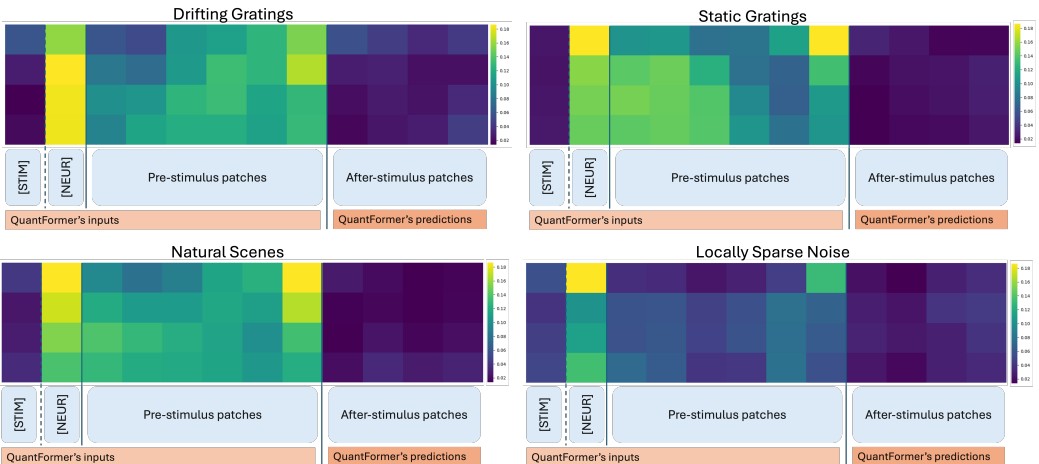

Figure A-8: Attention maps for all stimuli type.

# VII    INTERPRETABILITY OF LEARNED CODES

Fig. A-9 shows the latent space structure of discrete codes learned by vector quantization. We performed 2D t-SNE on a learned codebook to observe sequence patterns. Subfigure (a) shows that amplitude increases along the x-axis when plotting codes on the same scale. Subfigure (b) reveals pattern variability after normalizing the scale. Interestingly, despite having a relatively small number of codes, the reconstructed representation heavily depends on the sequence, as shown in Subfigure (c): we generated sequences with bursts of the same code, except for one typically representing a peak (e.g., code 19), highlighted between red dashed lines. The replaced code's amplitude and shape vary based on context, indicating that while codes represent patterns, the reconstruction depends on the whole sequence.

# VIII    INTERPRETABILITY OF NEURON EMBEDDINGS

To undestrand what is encoded into neuron embeddings, we visualized through t-SNE neuron embeddings from a downstream task. We find that neuron embeddings encode information such as activation frequency and response statistics. Colors in Fig. A-10 denote whether the measured quantity is above or below a threshold.

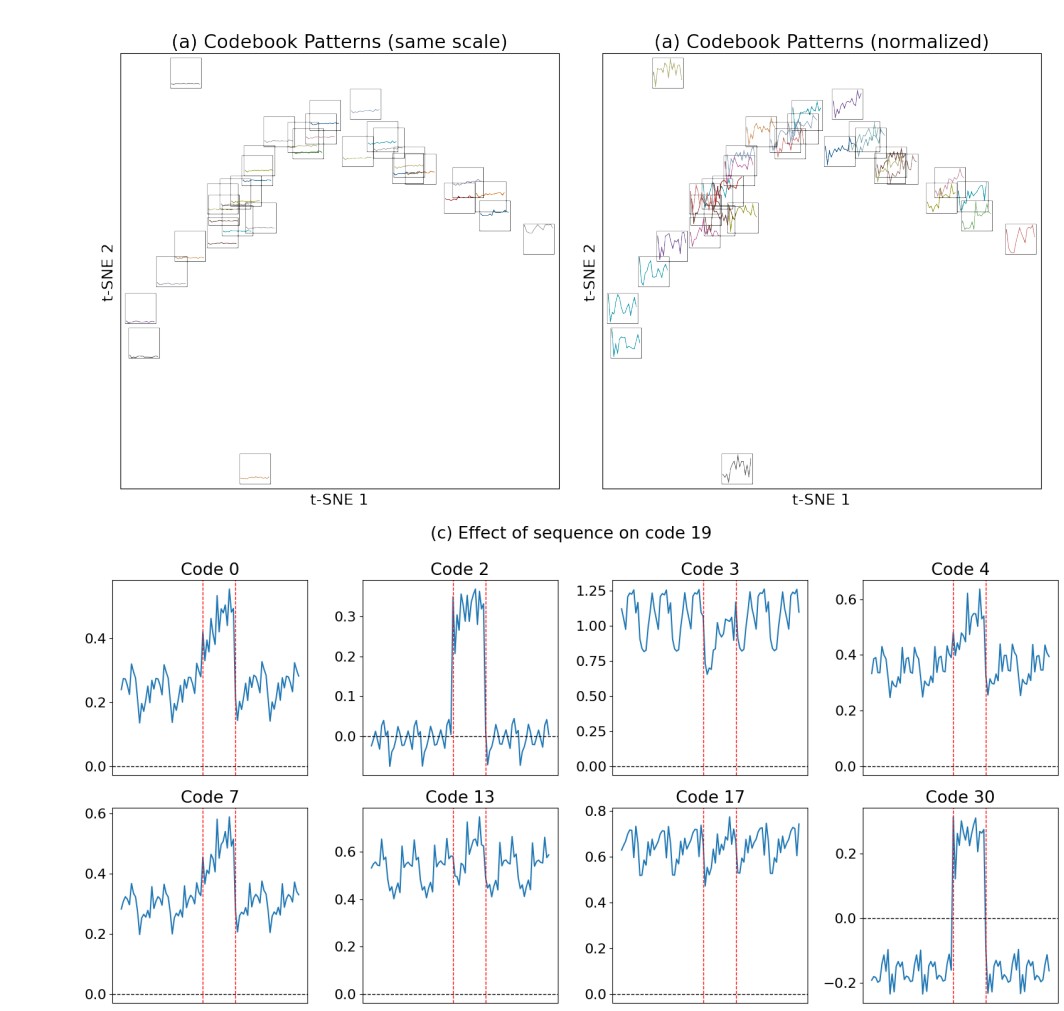

Figure A-9: Interpretability of codes. (a) t-SNE of a codebook, with patterns representation in the same scale. We can appreciate along the first axis the amplitude variation. (b) Same as before, but with normalization to appreciate differences in patterns. (c) Effect of sequence.

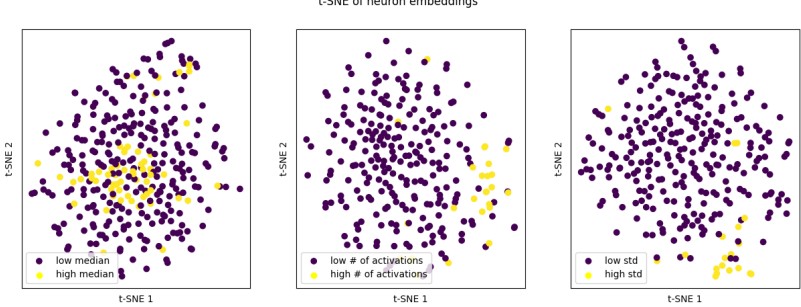

Figure A-10: Interpretability of neuron embeddings. We show t-SNE examples of neuron embeddings. We found that similar neurons in the latent space have also similar statistics like the median, the number of activations or the standard deviation.

