# OpenReview forum: "QuantFormer: Learning to quantize for neural activity forecasting in mouse visual cortex"
_ICLR.cc/2025/Conference — ICLR 2025 Conference Withdrawn Submission_

### Official Review · Reviewer_Tdst · 2024-10-15

**Soundness:** 2
**Presentation:** 1
**Contribution:** 2
**Rating:** 3
**Confidence:** 4

**Summary:**

The paper suggests to use a transformer to forecast neuronal activity with a focus on mice 2 photon imaged neurons. To tackle the sparse responses, the authors suggest to add quantization and classification loss derived from the quantization. They also try to tackle generalization issue and interpret their learnt neuron tokens.

**Strengths:**

Relatively original but not very clear paper, with interesting results.
I specifically liked
* A novel technical approach and an interesting idea with quantization, which seems to improve the results.
* The qualitative analysis of cross-former in comparison with Quant-Former (A6-A7)
* The authors made the first steps towards model interpretability, mainly in appendix, which is important for biological research.
* I appreciate attaching the code, this is a huge plus for reproducability

**Weaknesses:**

* Incomplete literature review and missing baselines
   * The Zhang et al 2024 work is not mentioned (https://arxiv.org/pdf/2407.14668 ) This work does neuron based forecasting but on neuropixels data.
   * The older works for neuronal forecasting, such as Zhu et al 2022 (https://www.nature.com/articles/s41593-022-01189-0) or Ye & Pandarinath 2021(https://arxiv.org/abs/2108.01210)  are not mentioned and also not used for baselines (Zhu et al 2022 is for calcium data).

While this models do not have stimuli tokens, one of them could still be a competitive baseline. I would also be interested to see the ablation showing the importance of the stimuli tokens for QuantFormer (is it about specific stimuli or stimuli type? )

* Incorrect statement about other works and incorrect citations
   * line 185  `SENSORIUM (Wang et al., 2023))`. Wang et al., 2023 does not use data from either Sensorium 2022 or 2023 competitions. It also barely discuss the competition
   * moreover, both SENSORIUM 2022 and 2023 provide spike traces data
   * For example, lines 176-177 *all the encoding and decoding methods discussed above rely on spiking data*, while in the mentioned works, Wang et al., 2023 (cited incorrectly though), Sinz et al 2019, Antoniades et al 2023, Turishcheva et al 2024 a/b all use calcium traces.
   * Same for lines 90-91, both Turishcheva et al 2024 a, and Microns (https://www.microns-explorer.org/cortical-mm3) provide open access to extensive datasets with calcium traces, not spikes.
   * Lines 144-147 *Approaches such as Turishcheva et al. (2024a;b); Li et al. (2023); Xu et al. (2023a); Sinz et al. predict neural responses based on stimuli, but often rely on trial-averaged data and are not designed to forecast future neural activity on a single-trial basis without the use of synchronous behaviour variables, which are not accessible in online settings.* While, indeed these approaches do not do neuronal forecasting, at least three out of four mentioned papers do not rely on either repeats or behaviour for responses prediction, only on the visual stimuli. Adding behaviour indeed improves performance, while repeats are used only during evaluations.
   * Mentioned  Antoniades et al 2023 work could be used for neuronal forecasting as well

* The biological validity of the paper is not clear
    * For table 2 it is not clear what are the upper/lower bounds for the metrics, which makes it hard to interpret how good all of the models generally are, as the correlation upper bound could not be one due to significant noise in the biological data. I would inspire the authors to use repeated stimuli within the session and follow Wang et al., 2023 to estimate at least the correlation upper bound.
    * It is also not clear, if the model is actually able to reproduce linear-nonlinear phenomenas, which the neurons should be able to do (like here https://journals.plos.org/ploscompbiol/article?id=10.1371/journal.pcbi.1009028 )

* The writing clarity could be improved.
   * For example,  in section `4.1 DATASET` it would be nice to explicitly write the amount of unique neurons. If the neurons across sessions for the same mice did not repeated, then its 11*3*250 $\approx$ 8500. If I understood the appendix correctly, there were 250 neurons per session on average. If these were exactly same neurons across sessions, it's 11*250 $\approx$ 3000. This makes very different impression compared to the 230 000 traces, which might be understood on neurons.
   * Generalization experiment is not explained (see questions)

**Questions:**

* How exactly do you perform the generalization experiment? You trained on 10 mice and evaluated on the other one?
But how the neuron-specific tokens were trained then for the test mouse? Also, were this neurons involved during pretraining? If yes, that might compromise the generalizatibility measure, as the model has see this data. What are the generalization ability of other models, like cross-former? Also, how are these numbers averaged? I am also not sure if it is really a good idea to measure generalization training on moving gratings and predicting the static ones as for the neurons ignoring motion this would be very close stimuli.

* Figure 1 states that neuronal forecasting models should take stimuli as input bit based on Figure 2- the model does not take visual stimuli as input but rather the stimulus-specific learnable tokens. Are this token per stimuli or per stimuli category (aka natural images, gratings, etc)?

Minor -
* In lines 518-519 *2D t-SNE on neuron embeddings (Fig. A-10) revealed that the [NEURON] token encodes neuron-specific statistics like activation probability* - the t-sne plot actually does not separate low and high-activated neurons, especially on the first plot. I would inspire authors to revise this statement
* Lines 369-370 *However, we excluded natural movies, as isolating individual neuron responses is challenging, and spontaneous activity, as it is not stimulus-related.* But how do you isolate spontaneuos activity for other stimuli categories?

**Details Of Ethics Concerns:**

Incorrect statements about cited papers

---

> ### Author Response · Authors · 2024-11-25
>
> Thank you for your detailed and thoughtful feedback. We appreciate the opportunity to address your concerns and provide clarifications. Thank you for bringing the cited works to our attention. We will include Zhang et al. (2024), Zhu et al. (2022), and Ye & Pandarinath (2021) in the literature review to provide a more comprehensive context for our contributions. Regarding Zhu et al. (2022), which focuses on calcium imaging data, we will conduct a direct comparison to evaluate our model against this baseline. For the ablation study on the importance of stimulus tokens, we are currently running the comparison. In the meantime, we encourage you to refer to our interpretability analysis in the appendix, which provides insights into how the model utilizes stimulus tokens during forecasting.
>
> Thank you also for pointing out inaccuracies. We will rephrase the relevant sentences to correct and clarify these points. First, we acknowledge that some cited works use deconvolved calcium traces, which involve additional preprocessing compared to raw fluorescence traces. To the best of our knowledge, raw fluorescence data without deconvolution are not commonly used in these works. We are actively working to extend our approach to include deconvolved traces for a more direct comparison.
> However, to the best of our understanding, Antoniades et al. (2023) (Neuroformer) focuses exclusively on spike data, as outlined in the workflow section of their paper. While some approaches (e.g., Turishcheva et al. 2024a;b, Li et al. 2023, Xu et al. 2023a) do not rely on trial-averaged responses during training, they do use repeats during evaluation or focus solely on visual stimuli for prediction. The total number of unique neurons used in our work is 3,989, considering the partial overlap between sessions. The mentioned 230,000 traces refers to the total number of trials, not neurons. We will ensure the work accurately reflects existing contributions and highlights our approach's unique focus on trial-specific neural forecasting with minimal preprocessing and we will correct this terminology to avoid any misunderstanding.
>
> Regarding your questions:
> For subject generalization, we pretrained the model on data from 10 mice and fine-tuned it on the 11th mouse. During fine-tuning, neuron-specific information was incorporated to adapt the model to the test mouse. Importantly, the neurons used during the fine-tuning phase were not included in the pretraining phase. For stimulus generalization, we agree that static and moving gratings share similarities, particularly for neurons that are less sensitive to motion. However, the conditions still differ because parameters like orientation and spatial frequency vary, creating a meaningful distinction for evaluating generalization. All results were averaged over 44 runs, representing fine-tuning experiments across different combinations of mice and stimulus types.
> The STIM token is derived from a representation of each stimulus, and what is learned is a linear projection that maps stimulus features to the embedding size. Specifically, we employ the following stimulus features:
>   - Natural Scenes: We use the features from the last layer of a ResNet-50 as input.
>   - Gratings: We encode information about spatial frequency, temporal frequency, orientation, and phase into four real-valued numbers.
>   - Locally Sparse Noise: We encode the \(x, y\) positions of white and black points into a vector. For gratings and locally sparse noise, we prefer our encoding method because ResNet-50 is pretrained on natural images, and thus, its representation of these stimuli most probably will not be significant.

---

> ### Comment · Reviewer_Tdst · 2024-11-26
>
> Thanks a lot for clarifications.
> I have one more question
>
> > For subject generalization, we pretrained the model on data from 10 mice and fine-tuned it on the 11th mouse
>
> Did you freeze some parts of the model during fine-tuning?

---

> > ### Author Response · Authors · 2024-11-26
> >
> > No, we conducted full fine-tuning on the encoder component.
> > However, it is important to note that the decoder (see Fig. 2 in the paper) was not re-trained, as its sole purpose is to reconstruct the signal.

---

> > > ### Comment · Reviewer_Tdst · 2024-11-26
> > >
> > > Thanks!

---

### Official Review · Reviewer_U5z1 · 2024-10-31

**Soundness:** 3
**Presentation:** 3
**Contribution:** 1
**Rating:** 3
**Confidence:** 4

**Summary:**

This paper introduces QuantFormer, a novel transformer-based model designed to forecast neural activity in mouse visual cortex using two-photon calcium imaging data. The authors reframe neural forecasting as a classification problem through vector quantization, and employ neuron-specific tokens to identify single neurons. The approach uses a two-stage training process combining pre-training through masked auto-encoding with downstream training for neural activity classification and forecasting. The network is trained on raw fluorescence traces from the Allen dataset traces rather than spike data or deconvoled fluorescence data.

**Strengths:**

- Very paper is well written. Previous literature and related work is mostly addressed, although a few citations might be missing.
- The main novelty seems to be the quantization stage that can predict the activity of neurons as a function of 32 codebook vectors.

**Weaknesses:**

- A closer examination of QuantFormer's architecture raises important questions about its broader applicability and evaluation methodology. The encoder-quantizer-decoder architecture, while novel, relies on a relatively small codebook of 32 entries to represent all possible neuronal activity patterns. Since the codebook and decoder remain frozen after pre-training (as far as I understand), this potentially limits the model's ability to represent novel activity patterns not seen during pre-training. This yields a model that is strictly not image computable and has therefore less abilities than a simple CNN model based on images (which could be trained to forecast as well). This is a major limitation of the model that is not properly discussed.
- As a direct result from using a finite set of stimuli, also the training and test protocol raises questions. As far as I understand, the pre-training phase includes data from the same neurons and images that appear in the test set, which may allow the model to learn specific response patterns before the actual testing phase. Because the Allen datasets includes repetitions of stimuli, this could be a major confounder for the results as the model can simply learn the mean responses given the the stimulus. The authors describe that they do not use neuron identities for pretraining. However, I (a) find this a questionable choice because this should severely limit the prediction capabilities of the model (what’s common to all neurons in cortex?) and (b) I am still not convinced that this would avoid the problem that the model learns mean responses of neurons.
- Because of that, I find the contributions overstated:
    - Forecasting for optogenetic manipulations is mentioned I could not find any experiment on that.
    - Forecasting has been addressed by other transformer architecture, for instance the “universal translator” by Zhang et al (see below), which is not cited or compared to as far as I can see.
    - Reframing forecasting of time series as a classification problem is per-se not a contribution if it doesn’t solve problems. As argued above, it seems to create problems.
    - Handling arbitrary neuronal populations is not new as other works (such as the POYO model) already use neuron ID tokens.
    - Finally, I would hardly call a model that is trained to forecast neurons from a finite set of stimuli a “foundation model for visual cortex”. In particular, I would expect a foundation model to be image/video computable.
- I find the choice of dataset not well motivated. The authors argue with real-time applicability. But then they don’t test it in those conditions. So in that sense they could apply it to a bigger dataset such as SENSORIUM 2022 (if they still want to exclude videos). My guess is that the method will not work well as it contains many unique stimuli in the training set. In particular, the choice of dataset is at odds with the motivation for the codebook (sparsity). I would expect fluorescence data to be less sparse than deconvolved data (such as Sensorium). It that sense SENSORIUM or spiking data should be even better data. Finally, I do not understand how they can get baseline activity for neurons in the Allen data. As far as I remember about the dataset, images are presented back to back. This means that neuronal firing does not return to baseline between images. I do not see how this is addressed in the paper.
- I find the choice of models to compare to a bit weak. I would recommend to include at least an oracle estimator that uses the mean responses of the neuron to that stimulus in the training set. Additionally, my guess is that a model pretrained properly on SENSORIUM and then trained to forecast a fixed number of steps in the future, should be competitive.
- Why is neuronal identity and the stimulus ignored during pretraining? I do not understand the rationale for it. Why is it not trained on forecasting with neuronal activities since this seems to generate problems (as discussed in 3.4.2)
- I find motivation for the classification into active and non-active not clear. It somehow assigns a special role of 10% more activity, which seems arbitrary. It also raises a question how the baseline is computed if the images are shown back to back (see above).
- I do not find the evaluation metrics very clear. How are correlations computed (across what and are correlations averaged over).
- In the appendix, the authors show a table with forecasting of unnormalized responses. The scores there are much lower. I do not find the explanations of the authors very clear here. I think this raises a question whether the normalization scheme somehow favors some models. Maybe a visual comparison of of normalized vs. non-normalized responses would help. Or a more detailed motivation for why normalization by the accumulated gradient helps. Also, I do not find this very clear (What gradient? Accumulated over what? Isn’t an accumulated gradient equal to the original signal up to a constant?).

**Minor weaknesses (I assume you will handle those, no need to respond to them):**

- `citep` and `citet` not consistently used. Please double check to use `citet` for inline citations and `citep` else.
- I would not call deconvolved calcium signals “spiking activity”. For instance, Turishcheva et al. does not use spikes, but deconvolved Ca++ activity.
- Possibly additional work to cite for forecasting
    - Zhang, Y., Wang, Y., Jimenez-Beneto, D., Wang, Z., Azabou, M., Richards, B., Winter, O., International Brain Laboratory, Dyer, E., Paninski, L., & Hurwitz, C. (2024). Towards a “universal translator” for neural dynamics at single-cell, single-spike resolution. In arXiv [q-bio.NC]. arXiv. Retrieved from http://arxiv.org/abs/2407.14668
    - Schmidt, F., Shrinivasan, S., Turishcheva, P., & Sinz, F. H. (2024). Modeling dynamic neural activity by combining naturalistic video stimuli and stimulus-independent latent factors. In arXiv [q-bio.NC]. arXiv. Retrieved from http://arxiv.org/abs/2410.16136
- Possibly interesting work to cite for neuronal quantization
    - Wei, X.-X., Zhou, D., Grosmark, A., Ajabi, Z., Sparks, F., Zhou, P., Brandon, M., Losonczy, A., & Paninski, L. (2019). A zero-inflated gamma model for post-deconvolved calcium imaging traces.
- Wrong reference for SENSORIUM. The correct ones are (alternatively use the Retrospective papers from NeurIPS 2023 or NeurIPS 2024)
    - Turishcheva, P., Fahey, P. G., Hansel, L., Froebe, R., Ponder, K., Vystrčilová, M., Willeke, K. F., Bashiri, M., Wang, E., Ding, Z., Tolias, A. S., Sinz, F. H., & Ecker, A. S. (2023). The Dynamic Sensorium competition for predicting large-scale mouse visual cortex activity from videos. In arXiv [q-bio.NC]. arXiv. Retrieved from http://arxiv.org/abs/2305.19654
    - Willeke, K. F., Fahey, P. G., Bashiri, M., Pede, L., Burg, M. F., Blessing, C., Cadena, S. A., Ding, Z., Lurz, K.-K., Ponder, K., Muhammad, T., Patel, S. S., Ecker, A. S., Tolias, A. S., & Sinz, F. H. (2022). The Sensorium competition on predicting large-scale mouse primary visual cortex activity. In arXiv [q-bio.NC]. arXiv. Retrieved from http://arxiv.org/abs/2206.08666

**Questions:**

- Can you motivate a clear benefit of your choice of your architecture?
- Can you cleaner motivate the choice of the Allen dataset and how you avoid data leakage between training and test? Could you conduct an experiment with completely separate neurons and images in training and test?
- Related: Can you run your model on the SENSORIUM 2022 data to show that it can deal better with unique image IDs?
- Can you provide a better explanation for your normalization scheme and why it improves model performance?
- Can you define how baseline activity is measured given the back-to-back image presentation in the Allen dataset.
- Can you address the potential mismatch between the sparsity motivation and the use of fluorescence data rather than deconvolved or spiking data?
- Can you provide a more detailed explanation of your normalization method, including a clear definition of the "accumulated gradient" and how it is calculated?
Can you conduct an analysis of how the normalization scheme affects different models' performance, to ensure it's not unfairly advantaging certain approaches?
Can you provide a clearer justification why this particular normalization method was chosen?

---

> ### Author Response · Authors · 2024-11-25
>
> Thank you for your detailed and insightful feedback.
>
> As a general consideration, the choice of 32 codes results from careful experimentation, balancing effectiveness within our methodology. Increasing codes poses challenges akin to regression-based approaches, with most codes representing baseline activity, complicating classification.
>
> If "image computable" refers to image-based features, our tokenization process accommodates features from CNNs or low-level image features, allowing us to handle diverse stimuli.
> We appreciate your feedback regarding the finite set of stimuli and the potential overlap between training and testing data. While pretraining includes the same neurons and stimuli, the model captures trial-specific dynamics beyond average stimulus-driven activity. Preliminary evidence is shown in the supplementary material (Fig. R1), comparing our average response to the training average.
>
> Regarding neuron identities, not using them during pretraining encourages the model to focus on common features across neurons rather than neuron-specific patterns. This enables the model to learn generalizable patterns during pretraining. When neuron identities are later introduced during fine-tuning, the model combines general and neuron-specific information, achieving its full predictive capability.
>
> More specifically regarding your questions,
>
> 1. The clear benefit of our architecture lies in the use of quantization, which enables us to reformulate the forecasting problem as a classification task.
> 2. The Allen dataset was chosen because it provides fluorescence data making it a benchmark for studying neural dynamics in response to visual stimuli. Temporal leakage is carefully avoided by splitting session data along the temporal dimension: training and testing data from the same session are separated by a gap of at least 10 min, ensuring that the model cannot directly learn from overlapping temporal patterns within a session.
> 3. We cannot conduct experiments with completely separate neurons because our model relies on learning neuron identities, which are session-specific. Regarding different stimuli, we cannot separate stimuli into training and testing sets within the Allen dataset without shuffling data along the temporal dimension, which would introduce information leakage. For example, test responses might inadvertently be included in the training set as "baselines." This limitation arises from the construction of the Allen dataset, not from our approach. However, our method does not rely solely on simple stimulus IDs, but rather on stimulus features. For example, for natural scenes, we use features extracted by a pretrained ResNet, which are then projected into the model's d-dimensional space via a trainable linear layer. We will take your suggestion to evaluate the model on the SENSORIUM 2022 dataset, in future updates.
> 4. (and 7) Derivative-based normalization is particularly effective for sparse signals characterized by distinct peaks because it mathematically emphasizes the sharp changes associated with these peaks while preserving their essential features. For a sparse signal $ x(t) $ consisting of distinct peaks, the signal can be approximated as: $x(t) = \sum_i A_i \delta(t - t_i),$ where  $A_i$ represents the peak amplitudes and \( t_i \) are the peak locations. The derivative of this signal is given by: $x'(t) = \sum_i A_i \delta'(t - t_i),$ which highlights sharp changes at the peak locations due to the presence of the derivative of the Dirac delta function, $ \delta'(t) $. To normalize the signal using its derivative, we define: $x_{\text{norm}}(t) = \frac{x(t)}{\lvert x'(t) \rvert}.$ This normalization amplifies the regions with significant changes (i.e., the peaks) while diminishing other less significant parts of the signal.
> 5. For each trial, we calculate the baseline activity using a 2-second window before the stimulus onset. We then compute the ΔF/F (change in fluorescence) relative to this baseline, which allows us to normalize the responses. We consider a neuron to be active if its ΔF/F exceeds 10%.
> 6. We use sparsity in a broad sense, meaning that neural activity is rare compared to base activity, which aligns with the structure of fluorescence data. We chose not to use deconvolved traces or spiking data because these require additional preprocessing steps, and our goal is to avoid such steps to remain consistent with a real-time scenario. We agree that deconvolved traces are indeed more sparse and could potentially provide additional benefits in certain contexts, but our focus is on minimizing preprocessing to streamline applicability. That said, we will test our model on the SENSORIUM 2022 dataset to evaluate its performance under different conditions. Regarding spike data, their fundamentally different nature makes quantization less meaningful, as spiking activity is already discrete and does not benefit from this approach.

---

> > ### Comment · Reviewer_U5z1 · 2024-11-27
> > **I stand with my previous evaluation**
> >
> > Thanks for your feedback and the additional details. However, they did not address my fundamental concerns. While *using* quantification might be novel, I do not think it solves an existing problem in neuronal forecasting. On the contrary, I think reducing it to classification produces problems because of limitation to the finite codebook. However, because of the choice of data set, we cannot say that for sure. So until you provide evidence that you can also forecast with much bigger image sets where training and test do not overlap, I will go with the baseline assumption that reducing it to 32 vectors does not help.
> >
> > I find the real time argument for fluorescence data weak, because you don't test your model under real time conditions. In addition, since fluorescence data will be stronger correlated over time (because of the Ca indicator response kernel), forecasting should be easier. Thus, you will have to accept the concern that you picked an "easy" dataset for your problem. Finally, should the model be compared in real time conditions in the future, I think it needs a justification when experimenters would care about real time forecasting. At the moment, with the given datasets, I could just use the previous responses of the neuron to that stimulus, average them, and use that as forecasting into the future. This, then, would also be a good baseline to compare against.
> >
> > Clarification regarding the term "image computable": I would consider a model image computable, if you can present it with an unseen image at test time and the model produces a prediction for that.

---

> > > ### Author Response · Authors · 2024-12-02
> > >
> > > Thank you for your feedback.
> > >
> > > While it is true that training and test images (mostly) overlap, the trials themselves do not, as we are modeling single-trial responses. This ensures that our model learns and evaluates on distinct neural activity patterns, making this a reasonable choice for our setup. Regarding stimulus generalizability, we understand the concern, but we believe it is a minor issue. The extracted features from the encoder are either derived from an external pretrained model or encoded as continuous variables, which makes overfitting by the linear layer unlikely. Additionally, for stimuli such as locally sparse noise, the model is trained under conditions where each trial features a unique configuration of black and white dots, generated randomly, ensuring independence between train and test trials.
> > >
> > > It is also important to note that the codebook is unrelated to the stimuli and instead depends on the response dynamics. The codebook size was selected through hyperparameter tuning to optimize the model’s ability to forecast responses effectively while balancing complexity and performance.
> > >
> > > On our point of view, we do not think that fluorescence data represents an easy dataset. While calcium signals are indeed temporally correlated due to the calcium indicator response kernel, this also introduces significant noise and variability at the single-trial level, which makes forecasting more challenging.
> > >
> > > Regarding the real-time applicability, while our work does not explicitly test real-time conditions, the motivation is rooted in enabling single-trial inference, which is critical for experimental neuroscience contexts such as optogenetic manipulations. In such experiments, models that predict future neural activity can guide dynamic, real-time interventions, offering substantial advantages over methods relying on averaged responses, which fail to capture trial-specific variability.
> > >
> > > We acknowledge your point that testing the model on real-time conditions and larger datasets with non-overlapping training and testing images would strengthen the argument. We are working toward expanding our evaluation to address these concerns in future studies.

---

### Official Review · Reviewer_4NGY · 2024-11-02

**Soundness:** 2
**Presentation:** 3
**Contribution:** 2
**Rating:** 3
**Confidence:** 4

**Summary:**

The paper proposes a transformer-based architecture for single-neuron response forecasting. In a first step, the authors use autoencoding to pre-train an encoder or neuronal response sequences. In a second step, they fine-tune the encoder on an activity classification and a response forecasting task. They evaluate their model using visually evoked responses in the visual cortex of mice. Compared to a few forecasting models they achieve improved activity classification and response forecasting metrics.

**Strengths:**

- Novel architecture and interesting idea in principle
- Paper is well written and easy to follow
- Shows some ablation experiments to tease apart which components are important

**Weaknesses:**

- Doing response forecasting on visually evoked responses seems like an odd choice
- Unclear how strong the baselines are
- Simple baselines like PSTH or linear encoding models are missing

**Questions:**

While I like the overall modeling approach and thinks it’s sane in general, I am somewhat confused by the authors’ choice of evaluation using visually evoked responses, which leaves me with lots of question marks whether the model works and how well. In my opinion, it is simply impossible to deduce anything about the performance of this model from the evaluation presented by the authors, because it does not properly deal with the visual stimulus. Although I doubt it, it is possible that I’m missing something. I would therefore like the authors to answer the following:

 1. When response forecasting is the goal, why do you use visually evoked responses, where the response is primarily determined by the stimulus rather than by past or pre-stimulus activity? There is some discussion around using the model online in experiments for optogenentic manipulation, but this motivation is not clear to me. In an experiment you control the visual stimulus that is shown, so you could easily incorporate it.
 1. If you choose to evaluate on visually evoked responses, the null model that you would have to beat is to simply take the PSTH of the neuron in response to the stimulus. I understand that your hypothesis may be that neurons do more than just responding to the stimulus and your goal is to explain this additional “noise” — but to drive home this point you first need a convincing stimulus-response baseline, onto which you can add the forecasting component.
 1. It appears to me that your model is trying to squeeze the stimulus-driven response patterns into a combination of neuron id token and stimulus token. Can you provide evidence against my hypothesis? Have you quantified for what fraction of the neurons’ response variance (during x_f) the PSTH accounts? Does your forecasting model exceed this value? I doubt it, given the correlation of 0.33 in Table 2.
 1. If you do not want to model the stimulus-driven response via an encoding model (or the PSTH), you should evaluate on datasets that do not have such strong external drive.
 1. Can you comment on whether your model beats any of your baselines on the datasets on which they have been tested and reported by the original authors? Did you train them yourself on the Allen dataset?

---

> ### Author Response · Authors · 2024-11-25
>
> Thank you for your thoughtful and detailed feedback. Below, we address each of your concerns and clarify our approach and its motivations.
>
> 1. We agree with this observation. However, traditional encoding methods are designed to predict the average response, which does not capture variability or trial-specific dynamics. In contrast, our approach focuses on forecasting individual trial responses, which is crucial for applications like online experiments with optogenetic manipulation. In such scenarios, real-time predictions of neural responses can guide dynamic adjustments, even when the visual stimulus is controlled.
> 2. To demonstrate how our approach captures trial-specific dynamics beyond the stimulus-driven average response, we visually compare our forecasting model to a PSTH-based baseline. Specifically, we averaged the responses across all trials in the training set to create a baseline (train average). During evaluation, we compared this baseline to both the average response across test trials (test average) and our model's average prediction (Refer to Fig. R1 in supplementary material). This analysis highlights how our method accounts for variability beyond the stimulus-driven average response.
> 3. We address this point using the attention maps provided in the appendix. These maps reveal that while neuron identity is the most significant factor in determining the forecasted indices, the final predictions are also influenced by the context surrounding the onset. This demonstrates that the model does not rely solely on stimulus-driven patterns but also incorporates pre-stimulus and trial-specific variability. Additionally, although the correlation value of 0.33 in Table 2 may seem modest, it reflects the inherent challenge of capturing trial-specific variance, which extends beyond replicating the average response.
> 4. (5) We have not directly compared our model's performance to the baselines on the original datasets reported by their authors, as our approach is specifically tailored to work with fluorescence traces, which differ significantly from data used in these studies. For the Allen dataset, we trained the baseline models ourselves under identical conditions to ensure a fair comparison. While we cannot claim to outperform these baselines on their originally tested datasets, our results highlight the strengths of our method in modeling trial-specific dynamics in fluorescence data.

---

> > ### Comment · Reviewer_4NGY · 2024-11-27
> >
> > Thank you for your response. I remain unconvinced by the replies. Qualitative plots of responses (point 2) and attention maps (point 3) are hints at most, but not evidence. Regarding point 5, I don't see a major difference of raw and deconvolved calcium traces. They're both continuous-valued time series related by (approximately) a linear convolution step. It's not clear to me why a method that works on one would not work on the other. Given that you happily train their method on your data for comparison, it's not clear to me why you wouldn't be able to train your method on their data, too.

---

> > > ### Author Response · Authors · 2024-12-02
> > >
> > > Thank you for your feedback. We understand your concern regarding the reliance on qualitative plots and attention maps as evidence. While these provide valuable insights, we agree that they are preliminary indications and not definitive evidence.
> > >
> > > Regarding point 5, we agree that raw and deconvolved calcium traces share similarities, as both are continuous time series. However, they also differ in their nature, with raw traces being inherently noisier and requiring additional effort to disentangle meaningful patterns. To run a fair comparison on deconvolved data, we believe it would be necessary to pretrain and finetuning our model appropriately, which would require an amount of time beyond the scope of this work and of the current time-window. Moreover, our focus on raw traces is motivated by minimizing preprocessing to align with real-time applications.
> > >
> > > We believe our model should also work with deconvolved traces and could be tested on datasets like SENSORIUM in the future. However, for the current work, our primary goal has been to demonstrate the efficacy of the model on fluorescence data, which we believe is a valuable contribution on its own.

---

### Official Review · Reviewer_ULZ9 · 2024-11-02

**Soundness:** 1
**Presentation:** 2
**Contribution:** 1
**Rating:** 3
**Confidence:** 3

**Summary:**

This paper introduces a large-scale model pretrained on the Allen corpus, which includes calcium imaging spiking activity from the mouse visual cortex under various stimulus conditions. It presents a transformer that uses vector quantization to create a set of neural codebooks for forecasting spiking activity. This quantization approach was shown to be effective in neural activity prediction, outperforming other baseline time series forecasting models. Additionally, the paper demonstrates positive scaling results across different stimuli and individual subjects.

**Strengths:**

This paper attempts to tackle an important problem of building "foundation models" for neuroscience that can predict spiking activity and classify responses to stimuli.

**Weaknesses:**

1. While vector quantization has not previously been used to build neuroscience foundation models, the author did not provide sufficient justification for choosing this specific model architecture.
2. The paper proposes only two types of downstream tasks for evaluating the foundation model. A more comprehensive evaluation is needed to assess the model's generalizability across diverse downstream tasks.
3. The paper lacks a scaling analysis to evaluate how effective the proposed backbone is for developing a foundation model.
4. The foundation model backbone could benefit from more rigorous benchmarking against existing methods on self-supervised prediction of spiking activity.

**Questions:**

**Major:**
1. In the introduction and related work section, the author cites many other time series models but doesn’t clearly motivate the choice of vector quantization for this work. Is this because this architecture has not been applied to neuroscience before? Although the author attempts to motivate the model choice in Section 3.3, it would be good to clarify the motivation earlier in the paper. **Could the author elaborate on why vector quantization was used or provide interpretable analysis, similar to what is found in the Appendix regarding the neural codebook?** I’m looking for a deeper discussion of how ML tools can help us answer unaddressed neuroscience questions, rather than just presenting another ML method that hasn’t been applied in the field.
2. Instead of comparing this model to other time series transformers, this method could be better benchmarked against existing work on self-supervised prediction of neural activity, such as NDT1 [1] and MtM [2]. Both methods can be repurposed for causal prediction of calcium imaging spiking activity and use masked modeling. **Could the author include experiments that compare their model to at least one of these approaches?**
3. Regarding model architecture, why not directly predict activity using a linear layer after the transformer encoder, as done in BERT? What is the rationale for using quantization and additional parameters in the transformer decoder? **An ablation study could help show the advantages of using quantization.** While Table 3 includes an ablation comparing quantization to an autoencoder, it would be more informative to compare it to a transformer baseline without quantization.
4. In Figure 3, I’m curious why the other baselines performed so poorly in predicting the target. It seems that **the evaluation could be conducted more carefully and fairly against other methods**. **For qualitative analysis, could the authors provide single-neuron peri-stimulus time histograms (PSTH) and single-trial activity after subtracting the PSTHs?** This would help clarify whether the model is only capturing the average pattern in the data.

**Minor:**
1. The author includes a stimulus token for neural activity forecasting, but incorporating stimulus information can also be considered a form of neural encoding. What would happen if stimulus information were excluded?
2. In Equation (2), why is the loss computed on both masked and unmasked portions? What is the rationale for balancing these two components, and what advantages does this provide?
3. I find Section 3.4.1 confusing. The author states that “feeding neuron and stimulus identifiers to the encoder is a key aspect of the approach.” Could this be clarified? Why not use per-neuron tokens as in POYO [3]? Additionally, what is the dimension of $t_1, ..., t_P$?
4. In the experiment section, the author mentions allocating two sub-sessions for training and testing, with each sub-session separated by 10-15 minutes. This interval seems quite long, and I wonder how the author addressed non-stationarity and potential distribution shifts in the neural data. Has the distribution of the training and test data been visualized?
5. In lines 392-394, the author claims that “brain signals can be encoded with 32 indices.” It would be cool for the author to further interpret this finding. However, I only found ablation studies on the number of quantization indices and a visualization of the learned neural notebook in the appendix. Does the author have a hypothesis as to why 32 indices are optimal?

[1] Ye, J., & Pandarinath, C. (2021). Representation learning for neural population activity with Neural Data Transformers.

[2] Zhang, Y., Wang, Y., Jimenez-Beneto, D., Wang, Z., Azabou, M., Richards, B., ... & Hurwitz, C. (2024). Towards a" universal translator" for neural dynamics at single-cell, single-spike resolution.

[3] Azabou, M., Arora, V., Ganesh, V., Mao, X., Nachimuthu, S., Mendelson, M., ... & Dyer, E. (2024). A unified, scalable framework for neural population decoding.

---

> ### Author Response · Authors · 2024-11-25
>
> Thank you for the detailed and insightful feedback. We greatly appreciate the opportunity to address your concerns and clarify the motivations, methodology, and results presented in our work. Below, we provide detailed responses to your comments.
>
> 1. Vector quantization was chosen to address the critical challenge of data sparsity in neuroscience. VQ compresses neural data into discrete representations, enabling robust pattern discovery (as shown in Fig. A-9) while preserving essential temporal structure. This transformation allows us to reformulate the response forecasting problem as a classification task, which is more effective than traditional regression methods that struggle with sparsity. We acknowledge that the motivation for this choice may not have been sufficiently emphasized early in the paper, and we will clarify it in the introduction to avoid confusion.
> 2. We are currently working to evaluate comparisons with NDT1 and MtM. However, this comparison is not straightforward because these methods were designed for calcium imaging spiking activity, while our approach operates on raw fluorescence traces. Although this may seem like a minor modification, it is challenging because fluorescence traces are continuous and noisier compared to the discrete nature of spiking data. To address this, we are experimenting with using inferred spike probabilities as inputs to these methods and converting their predictions back into time series data. We will include these results in future updates.
> 3. We indeed included this analysis in the paper; please refer to lines 474–476. The baseline in Tab. 3 is our encoder backbone, a BERT-like transformer encoder, followed by a linear layer. This model was trained using cross-entropy loss for activity classification and mean squared error (MSE) for forecasting, offering a direct comparison to our approach.
> 4. The observed superior performance of our method is due to its ability to use quantized indices for reconstructing the original shape of the signal, which enables it to capture more detailed patterns. In contrast, other methods rely on regression-based losses that focus on modeling average activity patterns. Although we cannot compute PSTHs strictly (as we are not using spikes), we can approximate them by averaging all responses for each stimulus and using this average as a baseline for comparison at test time. We included in the supplementary materials (Fig. R1) qualitative examples showcasing both cases where our model succeeds and where it fails, providing a balanced view of its performance relative to the baselines.
>
> Minor:
> 1. If stimulus information were excluded, the model would rely solely on the temporal structure and patterns in the neural activity itself for forecasting, likely reducing its predictive accuracy, particularly for responses tightly linked to specific stimuli. However, we conducted interpretability analysis to examine the attention distribution during predictions (see Fig. A-8). While the stimulus token is important, it plays a relatively minor role compared to neuron identity and has a comparable influence to the neuron's state before the stimulus onset.
> 2. We asked ourselves this question during preliminary experiments. The rationale is that a latent representation is meaningful if it enables recovery of the original input. When the reconstruction loss is applied only to masked tokens, the representation of unmasked tokens becomes poor, as their primary role is limited to aiding in reconstructing the missing parts. Moreover, while we did not conduct an exhaustive ablation study due to time constraints, preliminary experiments indicated that using the combined loss led to improved performance.
> 3. The [NEURON] embedding represents the identity of each neuron, ensuring that the model captures neuron-specific dynamics. Unlike POYO, where each spike is tokenized as an event, our approach tokenizes trace patches rather than individual spikes. Regarding the dimensions, the neuron, stimulus, and patches are embedded into vectors of dimension d, which matches the model's embedding space.
> 4. We visualized training and test data distributions (find attached an example in the supplementary material, Fig. R2) and observed significant overlap. As fine-tuning is always performed within the same session, we ensure the model adapts to within-session variability.
> 5. We conducted ablation studies on the number of indices and found that the best performance was achieved with 32 codes. However, we are not suggesting that 32 is the optimal encoding size for neural information in general—this is specific to our methodology. The relatively small number reflects how, as the number of codes increases, many of them end up representing neutral conditions (flat activity), leaving only a few for peak patterns. This imbalance creates challenges in classification, as too many similar codes make the task more like a regression problem, reducing effectiveness.

---

> > ### Comment · Reviewer_ULZ9 · 2024-11-29
> > **Thank you**
> >
> > I would like to thank the author for their response. However, I believe the proposed method requires (1) clearer clarification of its novelty, and (2) a more in-depth evaluation and comparison with baseline methods before it can be accepted at ICLR. At this stage, the method has potential for improvement in the future. I hope my suggestions could help the author improve this paper.

---

### Author Response · Authors · 2024-11-25

We extend our thanks to the reviewers for their thoughtful and constructive feedback, which has significantly enhanced our understanding of potential improvements and has been instrumental in clarifying our work. Below, we address the key points raised in your collective comments.

We greatly appreciate the suggestion to evaluate our results using peri-stimulus time histograms (PSHT). While our approach does not rely on spiking data, we have approximated PSHTs by averaging responses across trials and included qualitative examples in the supplementary materials. These examples aim to provide a clearer perspective on how our model handles stimulus-driven variability, and we would like to further develop this direction in future iterations.

We acknowledge the importance of benchmarking against existing methods to contextualize our contributions. However, as noted, our approach focuses on raw fluorescence traces rather than deconvolved traces or spiking data, and the inherent complexity of this task makes direct comparisons challenging. Although we plan to incorporate evaluations using deconvolved traces for a more comprehensive analysis, we emphasize that our objective is to enable real-time neural response forecasting with minimal preprocessing and algorithmic dependencies. This focus on efficiency and practicality underpins our methodological choices and aligns with real-time experimental scenarios, where preprocessing steps like deconvolution may be impractical. As highlighted, we intentionally avoided deconvolution to streamline preprocessing and preserve the raw nature of fluorescence data. However, we agree that deconvolved traces could complement our approach, and we aim to incorporate these aspects in future evaluations. Similarly, while the Allen dataset provided a robust foundation for our analysis, we are exploring the SENSORIUM 2022 dataset to validate our model under broader conditions.

Once again, we thank the reviewers for their suggestions and constructive feedback. We have addressed specific comments in personal responses, and we hope this global overview provides clarity on the core aspects of our work.

---

### Note · Authors · 2024-12-02

**Comment:**

We appreciate the reviewers' time and effort in evaluating our work. While we understand and respect their point of view, we do not fully share it. We believe our work addresses important challenges in neuronal forecasting and provides contributions especially in modeling single-trial neural dynamics and leveraging fluorescence data in innovative ways.

Although we stand by the significance of our contribution, we also understand that the reviewers will remain unconvinced, and their perspective is unlikely to change. With this in mind, we have decided to withdraw the paper at this time. The feedback provided will help us improve and strengthen our work for future submissions.

**Withdrawal Confirmation:**

I have read and agree with the venue's withdrawal policy on behalf of myself and my co-authors.